# Transcriptional responses in a mouse model of silicone wire embolization induced acute retinal artery ischemia and reperfusion

Yuedan Wang[1†], Ying Li[1†], Jiaqing Feng[1†], Chuansen Wang[1], Yuwei Wan[1], Bingyang Lv[1], Yinming Li[2], Hao Xie[2], Ting Chen[1], Faxi Wang[2], Ziyue Li[1], Anhuai Yang[1*], Xuan Xiao[1,2*]

[1]Department of Ophthalmology, Renmin Hospital of Wuhan University, Wuhan, China; [2]Department of Clinical Laboratory, Institute of Translational Medicine, Renmin Hospital of Wuhan University, Wuhan, China

*For correspondence:
yah0525@126.com (AY);
xiaoxuan1111@whu.edu.cn (XX)

†These authors contributed equally to this work

Competing interest: The authors declare that no competing interests exist.

**Abstract** Acute retinal ischemia and ischemia-reperfusion injury are the primary causes of retinal neural cell death and vision loss in retinal artery occlusion (RAO). The absence of an accurate mouse model for simulating the retinal ischemic process has hindered progress in developing neuroprotective agents for RAO. We developed a unilateral pterygopalatine ophthalmic artery occlusion (UPOAO) mouse model using silicone wire embolization combined with carotid artery ligation. The survival of retinal ganglion cells and visual function were evaluated to determine the duration of ischemia. Immunofluorescence staining, optical coherence tomography, and haematoxylin and eosin staining were utilized to assess changes in major neural cell classes and retinal structure degeneration at two reperfusion durations. Transcriptomics was employed to investigate alterations in the pathological process of UPOAO following ischemia and reperfusion, highlighting transcriptomic differences between UPOAO and other retinal ischemia-reperfusion models. The UPOAO model successfully replicated the acute interruption of retinal blood supply observed in RAO. 60 min of Ischemia led to significant loss of major retinal neural cells and visual function impairment. Notable thinning of the inner retinal layer, especially the ganglion cell layer, was evident post-UPOAO. Temporal transcriptome analysis revealed various pathophysiological processes related to immune cell migration, oxidative stress, and immune inflammation during the non-reperfusion and reperfusion periods. A pronounced increase in microglia within the retina and peripheral leukocytes accessing the retina was observed during reperfusion periods. Comparison of differentially expressed genes (DEGs) between the UPOAO and high intraocular pressure models revealed specific enrichments in lipid and steroid metabolism-related genes in the UPOAO model. The UPOAO model emerges as a novel tool for screening pathogenic genes and promoting further therapeutic research in RAO.

## eLife assessment

The manuscript establishes a sophisticated mouse model for acute retinal artery occlusion (RAO) by combining unilateral pterygopalatine ophthalmic artery occlusion (UPOAO) with a silicone wire embolus and carotid artery ligation, generating ischemia-reperfusion injury upon removal of the embolus. This clinically relevant model is **useful** for studying the cellular and molecular mechanisms of RAO. The data overall are **solid**, presenting a novel tool for screening pathogenic genes and promoting further therapeutic research in RAO.

## Introduction

RAO is a severe ophthalmic disease characterized by a sudden interruption of blood flow in the retinal artery, leading to retinal ischemia (*Scott et al., 2020*). Over 60% of RAO patients suffer from impaired vision, ranging from finger counting to complete vision loss (*Hayreh and Zimmerman, 2005*). Additionally, RAO patients face an increased risk of cardiovascular and cerebrovascular events (*Vestergaard et al., 2021*; *Fallico et al., 2020*; *Chen et al., 2023*). Retinal ischemia and hypoxia can lead to irreversible damage to retinal cells within 90 min (*Prasad et al., 2010*). This damage is primarily due to the apoptosis of retinal ganglion cells (RGCs), which is driven by inflammation and oxidative stress. Unfortunately, conservative treatments for RAO, such as ocular massage and hyperbaric oxygen therapy, provide limited therapeutic benefits (*Mac Grory et al., 2021*). Although thrombolysis is effective, its application is restricted by a narrow therapeutic window (typically within 4.5 hr *Raber et al., 2023*). Furthermore, the restoration of blood flow to the ischemic area following thrombolysis can cause a sudden increase in tissue oxidative levels, resulting in ischemia-reperfusion injury (IRI) in the retina (*Hu et al., 2022*). IRI is a common pathological condition that can induce inflammation, retinal tissue damage, and visual impairment (*Wan et al., 2020*). Therefore, it is crucial to develop an animal model that accurately simulates the pathological processes of RAO to thoroughly investigate the pathophysiological changes and explore potential neuroprotective treatments following retinal ischemia and reperfusion.

Various animal models have been employed to investigate the effects of retinal injury resulting from ischemia and subsequent reperfusion in RAO (*Prasad et al., 2010*; *Shabanzadeh et al., 2018*; *Gao et al., 2020*). Based on the methods used for modeling, retinal ischemia-reperfusion models can be categorized into two groups: (1) intravascular occlusion models, including photochemical-induced thrombosis model and vascular intervention model; and (2) extravascular occlusion models, including central retinal artery ligation (CRAL) model, unilateral common carotid artery occlusion (UCCAO) model, and high intraocular pressure (HIOP) model. The photochemical-induced thrombosis model and vascular intervention model have been reported to be useful tools for assessing retinal injury after ischemia and reperfusion. However, both models are limited by the need for trained interventional radiologists and the need for advanced techniques (*Morén et al., 2009*; *Zhang et al., 2005*). The CRAL model, which directly clips or ligates the central retinal artery, also presents drawbacks, such as potential optic nerve damage and difficulty in application to small animals such as mice, limiting its use in experiments with larger sample sizes (*Prasad et al., 2010*). The UCCAO model, which induces retinal hypoperfusion by occluding the unilateral common carotid artery (CCA), is more suitable for simulating chronic ischemic retinal disease than acute retinal ischemia (*Lee et al., 2022*; *Lee et al., 2022*). The HIOP model is widely utilized in mice for studying ischemia-reperfusion in acute primary angle-closure glaucoma (APACG *van Zyl et al., 2020*). However, the mechanical compression of the retina caused by saline injection in this model may also lead to retinal damage, potentially influencing IRI research. Therefore, it is critically important to develop a simple and low-skill-required mouse model that can simulate acute retinal ischemia and reperfusion injury in RAO patients.

To better simulate the retinal ischemic process and possible IRI following RAO, we developed a novel vascular-associated mouse model called the UPOAO model. In this model, we employed silicone wire embolization and carotid artery ligation to completely block the blood supply to the retina. We characterized the major classes of retinal neural cells and evaluated visual function following different durations of ischemia (30 min and 60 min) and reperfusion (3 days and 7 days) after UPOAO. Additionally, we utilized transcriptomics to investigate the transcriptional changes and elucidate the pathophysiological process of the UPOAO model after ischemia and reperfusion. Finally, we highlighted the transcriptomic differences between this model and other retinal ischemia-reperfusion models, including HIOP and UCCAO, revealing the unique pathological processes that closely resemble retinal IRI in RAO. The UPOAO model offers new insights into the mechanisms and pathways involved in ischemia and reperfusion studies of RAO, providing a foundation for studying protective strategies for ocular ischemic diseases.

## Results

### Silicone wire embolus insertion interrupts retinal blood flow

To reproduce the retinal ischemic process and potential IRI in RAO, we established a mouse model of UPOAO by combining silicone wire embolization with carotid artery ligation. Model validation was conducted through a series of experiments, with detailed descriptions provided in the Materials and methods section (*Figure 1A–G*). To confirm blood flow disruption, we performed cardiac perfusion with fluorescently labeled lectin and FFA. Rhodamine-labeled canavalin A was used during in vivo perfusion with silicone wire embolus, and both eyes were evaluated. The sham eye exhibited normal blood perfusion in the retina, while the experimental eye showed no perfusion (*Figure 1H, I*). FFA revealed delayed and limited perfusion in the experimental lateral retina, primarily near the optic disc (*Figure 1J and K*). These findings indicate that the insertion of the silicone wire embolus effectively impaired blood flow to the retina.

### 60-min ischemia in UPOAO damage retinal structure and function

To investigate the optimal ischemic duration, the mice were subjected to ischemic periods of 30 and 60 min, followed by reperfusion periods of 3 days and 7 days. Retinal structure and visual function were assessed through flat-mounted retina analysis and flash ERG.

In the 30 min ischemia group, no significant RGCs death was observed after 3 days (*Figure 2A and B*) or 7 days of reperfusion (*Figure 2C and D*). However, after 60 minutes of ischemia followed by either 3 days or 7 days of reperfusion, reductions in RGCs density were evident (*Figure 2E–H*). ERG results showed no statistical difference in amplitudes between bilateral eyes after 3- and 7 days of 30 min reperfusion (*Figure 3A–C*). However, the b-wave amplitude notably decreased after 60 min of ischemia and 3 days of reperfusion (*Figure 3D*). By 7 days of reperfusion, the b-wave amplitude had halved compared to that of the sham eyes (*Figure 3E*). The appearance times of the a-waves and b-waves in the experimental and control eyes remained consistent across all four groups (*Figure 3—figure supplement 1*). Additionally, OPs have also been used to sensitively monitor retinal ischemic effects by detecting changes before any alterations in b-waves occur (*Cao et al., 1993*; *Xu et al., 2022*; *Takács et al., 2024*). The amplitudes of OPs in the experimental eyes, particularly in the 60 min ischemia and 7 days reperfusion group, significantly decreased to less than 50% of those in the sham eyes (*Figure 3F and G*). Based on RGCs survival and changes in ERG waves, we determined that a 60 min ischemic duration is optimal for the UPOAO model.

### Evaluation of retinal thickness in UPOAO through OCT and HE

To assess retinal thickness non-invasively in the UPOAO model, we conducted OCT imaging during 3 days and 7 days reperfusion periods. Following a 3 days reperfusion period, no significant changes were observed in the GCC, INL + OPL, and ONL +IS/OS + RPE layers at 1.5, 3, and 4.5 PD from the optic disc (*Figure 4B*, *Figure 4—figure supplement 1A, B*). However, the total retinal thickness decreased at 4.5 PD (*Figure 4C*). After 7 days of reperfusion, the total retinal thickness decreased at 1.5 PD, 3 PD, and 4.5 PD, primarily due to inner retinal thinning (*Figure 4D–F*, *Figure 4—figure supplement 1C, D*).

For a detailed analysis of inner retinal thickness, we extracted UPOAO mouse eyes and stained them with HE. The thickness of the RNFL + GCL increased at 3 days of reperfusion (*Figure 4H*), followed by a decrease at 7 days of reperfusion (*Figure 4L*). IPL thickness decreased at both 3 days and 7 days of reperfusion (*Figure 4I and M*), while INL thickness decreased only at 7 days of reperfusion (*Figure 4N*). We hypothesized that the initial swelling of the GCL in response to acute ischemia and hypoxia led to early thickening, followed by gradual thinning due to tissue dysfunction. This hypothesis was supported by the observation of RGCs loss and reduced GCC thickness evaluated by OCT at 7 days (*Figure 4E*).

We observed by OCT that changes in retinal thickness became noticeable at 3 days post-UPOAO and exhibited a significant decrease at 7 days. HE results further revealed that alterations in inner retinal thickness constituted a substantial portion of the total retinal thickness during the early reperfusion stage after UPOAO.

### Survival of retinal neural cells in UPOAO

Given that mouse retinal function heavily relies on rod cells and that scotopic (low light) vision is primarily governed by rod photoreceptors, the observed reductions in b-wave and OPs-wave

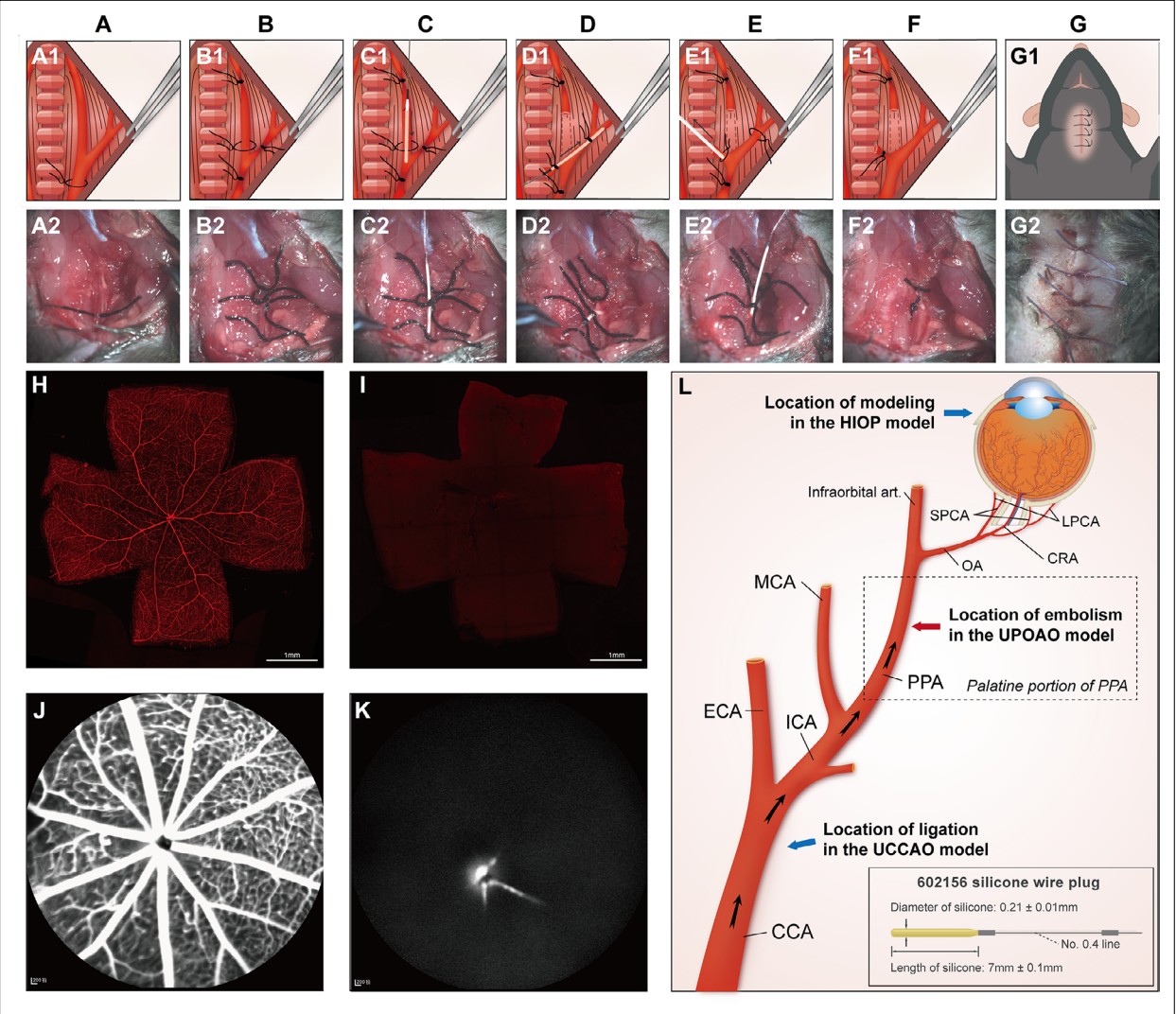

**Figure 1.** Modeling procedure, validation, and cervical artery anatomy. (**A1–G1**) Schematic illustration of unilateral pterygopalatine ophthalmic artery occlusion (UPOAO). (**A2–G2**) Practical operation of UPOAO. (**A1, A2**) Blunt separation and exposure of the left cervical arteries. (**B1, B2**) Arterial suture ligation. (**C1, C2**) Insertion of the silicone wire embolus. The artery was incised to create a hole, and the silicone wire embolus was inserted. (**D1, D2**) Artery disconnection and movement of the silicone wire embolus. The artery was cut along the incision, and the silicone wire embolus was retracted and reinserted. (**E1, E2**) Removal of the silicone wire embolus and reperfusion. (**F1, F2**) Suture removal. The sutures at both ends of the disconnected vessel were knotted, and the other two sutures were removed. (**G1, G2**) Anatomic reduction and suturing of the skin. (**H, I**) Canavalin A label vasculature of the UPOAO mouse retina. The silicone wire embolus was inserted into the artery before perfusing rhodamine-labeled canavalin A into the heart of the UPOAO mouse. The sham eye served as an unpracticed control eye, while the UPOAO lateral eye represented the experimental eye. Retinal vessels in the sham eye (**H**) exhibited fluorescence filling, while retinal vessels in the UPOAO lateral eye (**I**) remained unfilled. Scale bar = 1 mm. (**J and K**) Fluorescein fundus angiography (FFA) was performed before removing the silicone wire embolus from the UPOAO mouse. The vessels in the sham lateral retina (**J**) were perfused, while the lateral retinal perfusion in UPOAO (**K**) was delayed. (**L**) Schematic illustration of cervical artery anatomy and ocular blood supply. Embolization of the pterygopalatine artery (PPA) resulted in ocular ischemia. The red arrow indicates the site of the silicone wire embolus occlusion. The silicone wire embolus used a type 602156 wire, extended to 7 mm with a diameter of 0.21 mm. The blue arrows indicate the modeling locations of the high intraocular pressure (HIOP) model and the unilateral common carotid artery occlusion (UCCAO) model, respectively. CCA: common carotid artery; ICA: internal carotid artery; ECA: external carotid artery; PPA: pterygopalatine artery; MCA: middle cerebral artery; Infraorbital art.: infraorbital artery; OA: ophthalmic artery; SPCA: short posterior ciliary artery; LPCA: long posterior ciliary artery; CRA: central retinal artery.

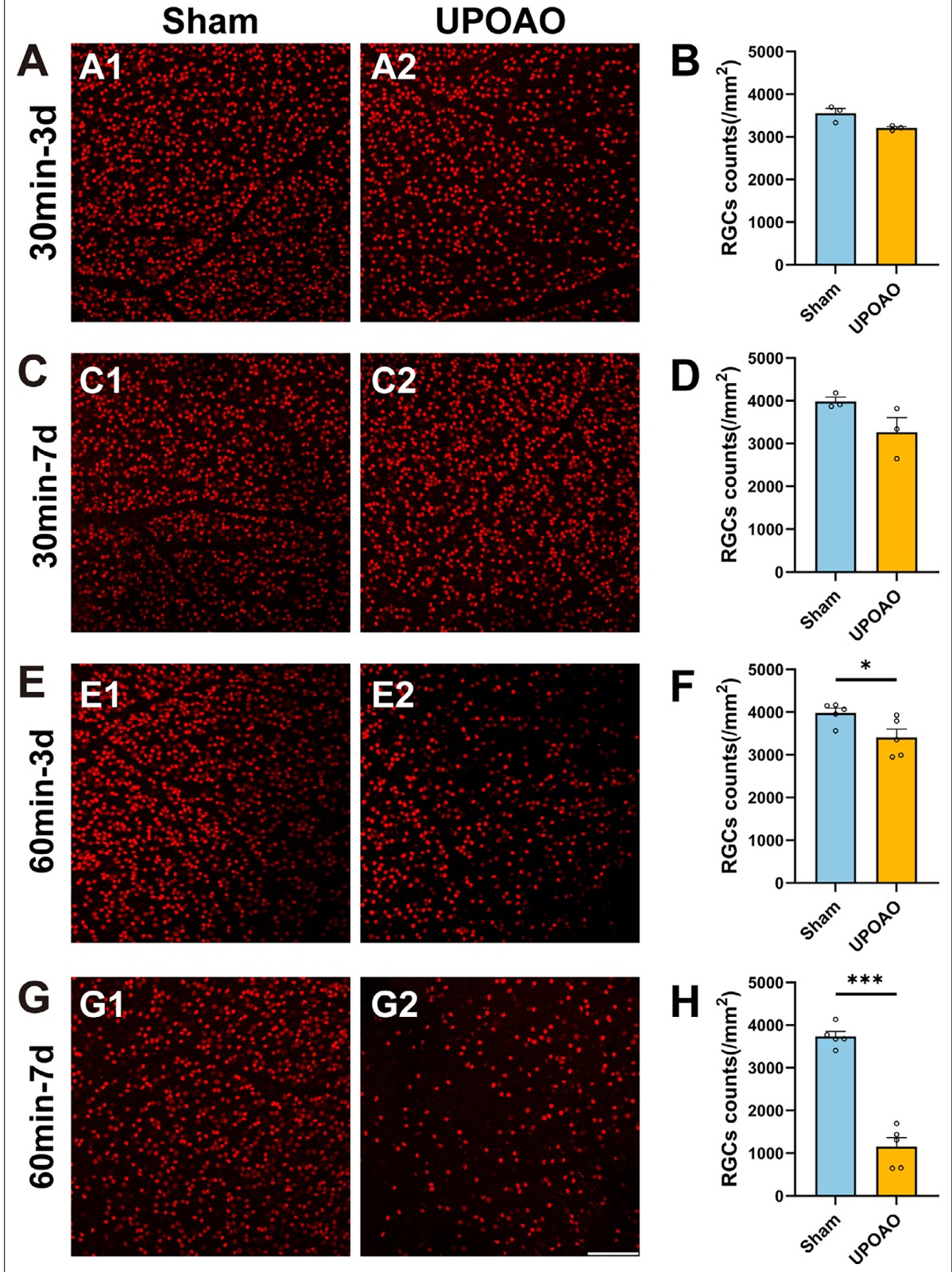

**Figure 2.** Staining and quantification of retinal ganglion cells (RGCs) at different ischemia and reperfusion times. Flat-mounted retina RGCs were labeled with Brn3a staining. (**A, B**) Representative pictures of the peripheral field (**A**), and quantification of surviving RGCs in all fields (**B**) in the 30 min ischemia and 3 days reperfusion group. n=3. (**C, D**) Representative pictures of the peripheral field (**C**), and quantification of surviving RGCs in all fields (**D**) in the 30 min ischemia and 7 days reperfusion group. n=3. (**E, F**) Representative pictures of the peripheral field (**E**), and quantification of

*Figure 2 continued on next page*

*Figure 2 continued*

surviving RGCs in all fields (**F**) in the 60 min ischemia and 3 days reperfusion group. n=5. (**G, H**) Representative pictures of the peripheral field (**G**), and quantification of surviving RGCs in all fields (**H**) in the 60 min ischemia and 7 days reperfusion group. n=5. The results showed a significant loss of RGCs after 60 min of ischemia. Data were presented as means ± s.e.m, *p<0.05, **p<0.01, ***p<0.001, ****p<0.0001, t-test. Scale bar = 100 μm.

amplitudes indicate impaired synaptic transmission within the inner retina (*Yang et al., 2019*). Our OCT and HE findings suggested structural injuries within the inner retina. Notably, the thickness of the outer retinal layers in OCT remained unchanged. To further explore these observations, we conducted immunofluorescence staining to investigate alterations in major retinal cell types, primarily focusing on bipolar cells (BCs), photoreceptor cells, horizontal cells (HCs), and cholinergic amacrine cells.

## BCs loss and photoreceptor cells survival

PKCα serves as a marker delineating BCs, with their cell bodies primarily located in the outermost part of the INL, axonal terminals extending into the innermost part of the IPL, and dendrites confined to the OPL (*Figure 5A and B*). In the experimental UPOAO eyes, BCs did not show significant changes at the 3 days reperfusion mark but exhibited dramatic alterations by the 7 days reperfusion period (*Figure 5A2 and B2*). In particular, the immunostaining density of BCs dendritic and axonal arbors notably decreased at the 7 days reperfusion mark. BCs account for approximately 40% of INL cells in mice, and their somata and axonal processes form a substantial part of the INL and IPL, consistent with the thinning of the INL observed in both OCT and HE at 7 days (*Figure 4E and N*). Notably, the decrease in the number of PKCα⁺ BCs at the 7 days reperfusion point coincided with the decrease in the b-waves amplitude in ERG (*Figure 3E*), indicating functional interplay between BCs and other cells.

Recoverin marks the somata and outer segments of photoreceptor cells and is localized within the ONL (*Figure 5—figure supplement 1A, B*). Interestingly, recoverin-positive photoreceptors remained relatively stable throughout the reperfusion periods, which aligns with the unchanged thickness of the outer retinal layers observed in OCT (*Figure 4—figure supplement 1C, D*).

## HCs and cholinergic amacrine cells loss

We evaluated HCs and cholinergic amacrine cells using calbindin and ChAT immunostaining, respectively. Calbindin immunostaining highlighted the somata of HCs within the INL, with terminal axon connections linearly positioned within the OPL (*Figure 5E and F*). In the UPOAO experimental eyes, no significant change in the number of HCs was observed during the 3 days reperfusion period, while a notable reduction was observed after 7 days. By 7 days, a significant decrease in cell body numbers, a considerable reduction in axon density, and disrupted linear connections were observed (*Figure 5F2*).

ChAT⁺ cell bodies were located in the GCL and INL, with dendrites forming two narrow stratified bands within the IPL. The immunofluorescence density of ChAT⁺ amacrine cells decreased notably after 3 days and even more prominently after 7 days (*Figure 5I and J*). The fluorescence intensity of the two bands within the IPL markedly decreased and was nearly invisible after 7 days (*Figure 5J2*).

## Time course transcriptome analysis revealed features of different reperfusion periods in UPOAO

To explore the pathophysiological processes of UPOAO, we extracted retinas from both eyes for transcriptome sequencing. The samples included retinas from eyes subjected to 60 min of ischemia without reperfusion, 60 min of ischemia followed by 3 days of reperfusion, and 60 min of ischemia followed by 7 days of reperfusion, with the sham-treated eyes serving as controls.

In the non-perfusion group, 215 genes were upregulated and 204 genes were downregulated (*Figure 6A*). GO enrichment analysis revealed that the DEGs were related to leukocyte migration, epidermis development, myeloid leukocyte migration and other cell migration pathways (*Figure 6B*). The heatmap and box showed the upregulated and downregulated genes involved in immune cell migration-related pathways during the non-perfusion period in UPOAO (*Figure 6C*). High-connectivity DEGs ('hub genes'), such as *Dusp1* and *Fos*, were also enriched in these pathways (*Figure 6—figure supplement 1A*). GSEA also showed similar results (*Figure 6—figure supplement 1B–D*). These

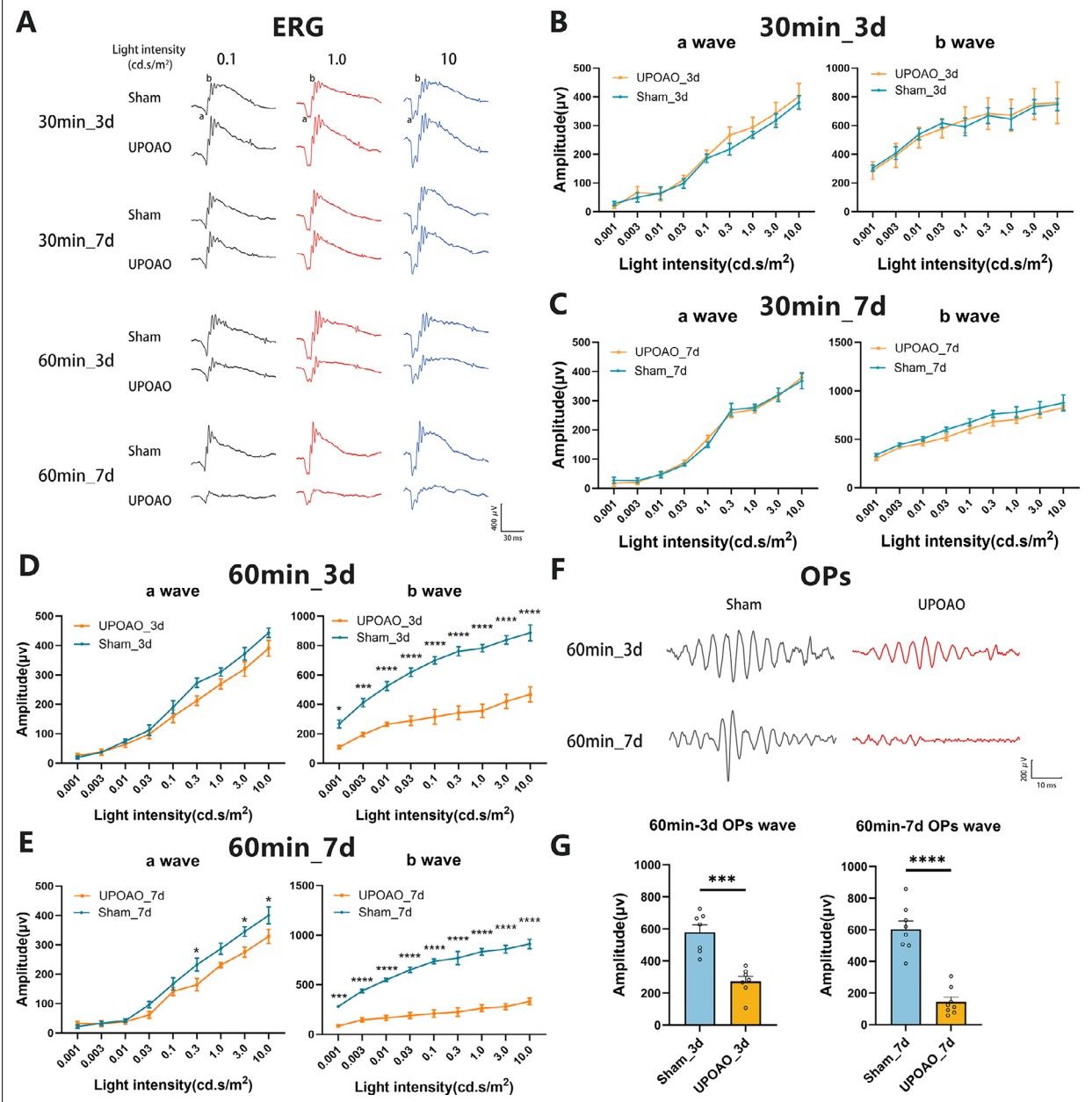

**Figure 3.** Comparison of electroretinographic (ERG) dark-adapted responses at different ischemia and reperfusion times. Following the evaluation of surviving retinal ganglion cells (RGCs), visual function in sham and unilateral pterygopalatine ophthalmic artery occlusion (UPOAO) experimental eyes at various ischemia and reperfusion times was assessed using ERG. (**A**) Representative waveforms in the four groups at the stimulus light intensities of 0.1, 1.0, and 10.0 cd.s/m2, respectively. (**B**) Quantification of a-wave and b-wave amplitudes in the 30 min ischemia and 3 days reperfusion group. n=5. (**C**) Quantification of a-wave and b-wave amplitudes in the 30 min ischemia and 7 days reperfusion group. n=5. (**D**) Quantification of a-wave and b-wave amplitudes in the 60 min ischemia and 3 days reperfusion group. n=7. (**E**) Quantification of a-wave and b-wave amplitudes in the 60 min ischemia and 7 days reperfusion group. n=8. Dark-adapted responses showed almost similar a-wave amplitudes but significantly decreased b-wave amplitudes in the 60 min ischemia groups. The amplitudes of b-waves declined at 3 days and even more prominently at 7 days. (**F, G**) Representative oscillatory potentials (OPs) and quantification of amplitudes in the 60 min ischemic groups. n=7 in the 3 days reperfusion group; n=8 in the 7 days reperfusion group. The amplitudes of OPs decreased significantly at 7 days reperfusion. The decline in b-waves and OPs along with the loss of RGCs, supports the selection of a 60 min ischemic duration as an appropriate choice. Data were presented as means ± SEM, *p<0.05, **p<0.01, ***p<0.001, ****p<0.0001, two-way ANOVA test for a-waves and b-waves; paired t-test for OPs.

The online version of this article includes the following figure supplement(s) for figure 3:

**Figure supplement 1.** Response times of a-waves and b-waves in electroretinographic (ERG) at different light intensities.

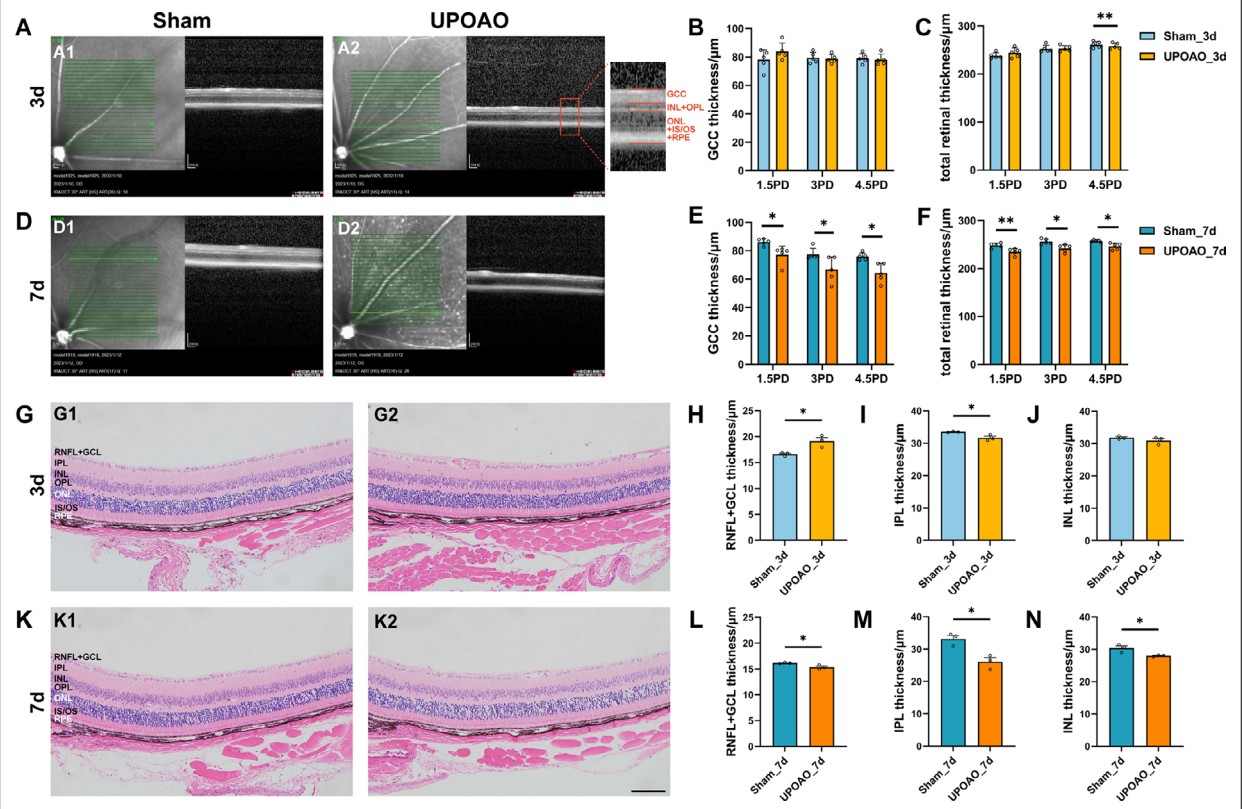

**Figure 4.** Changes in retina morphology in the unilateral pterygopalatine ophthalmic artery occlusion (UPOAO) model. (**A**) Representative optical coherence tomography (OCT) images of the mouse retina at 3 days. The Green lines indicate the OCT scan area starting from the optic disc. Local magnification and layering are annotated. ganglion cell complex (GCC): ganglion cell complex, including RNFL, GCL, and IPL layers. (**B and C**) Quantification of GCC and total retinal thickness at 3 days. The thickness of the GCC and the entire retina in OCT was measured and compared at distances of 1.5 PD, 3.0 PD, and 4.5 PD from the optic disc, respectively. n=5. (**D**) Representative OCT images of the mouse retina at 7 days. (**E and F**) Quantification of GCC and total retinal thickness at 7 days. n=5. (**G**) Representative hematoxylin and eosin (HE) images of the mouse retina at 3 days. (**H, I and J**) Quantification of nerve fibre layer (NFL) + GCL, IPL, and INL thickness at 3 days. Retinal thickness in HE was measured near the optic nerve head and compared. n=3. (**K**) Representative HE images of the mouse retina at 7 days. (**L, M and N**) Quantification of NFL + GCL, IPL, and INL thickness at 7 days. n=3. Data were presented as means ± s.e.m, *p<0.05, **p<0.01, ***p<0.001, ****p<0.0001, two-way analysis of variance (ANOVA) test in OCT and paired t-test in HE. PD: papillary diameters; RNFL: retinal nerve fiber layer; GCL: ganglion cell layer; IPL: inner plexiform layer; INL: inner nuclear layer; OPL: outer plexiform layer; ONL: outer nuclear layer; IS: inner segment; OS outer segment; RPE: retinal pigment epithelium. Scale bar = 100 µm.

The online version of this article includes the following figure supplement(s) for figure 4:

**Figure supplement 1.** Quantification of inner nuclear layer (INL) + outer plexiform layer (OPL) thickness and outer nuclear layer (ONL) + inner segment/outer segment (IS/OS) + retinal pigment epithelium (RPE) thickness in optical coherence tomography (OCT) during 3 days and 7 days Reperfusion of unilateral pterygopalatine ophthalmic artery occlusion (UPOAO) Animals.

results suggested that in the early stage of retinal ischemic injury, leukocytes from the microvasculature may infiltrate retinal tissue. More experimental validation will be performed to confirm this hypothesis.

In the 3 days reperfusion group, 372 genes were upregulated, and 170 genes were downregulated (*Figure 6D*). GO enrichment analysis revealed that the DEGs at 3 days after reperfusion were related to energy metabolism, mitochondrial regulation, and oxidative stress pathways (*Figure 6E*). The heatmap and box displayed the upregulated and downregulated genes involved in oxidative stress-related pathways at the 3 days reperfusion stage in UPOAO, including hub genes such as *Mrpl18* and *Mrps23* (*Figure 6F*, *Figure 6—figure supplement 2A*). GSEA revealed changes in pathways related to the binding membrane of organelles, the cell surface, and the plasma membrane protein complex (*Figure 6—figure supplement 2B*). Furthermore, 44 overlapping genes between the mitochondrial genes and DEGs at 3 days post-UPOAO were mainly related to mitochondrial transport and the

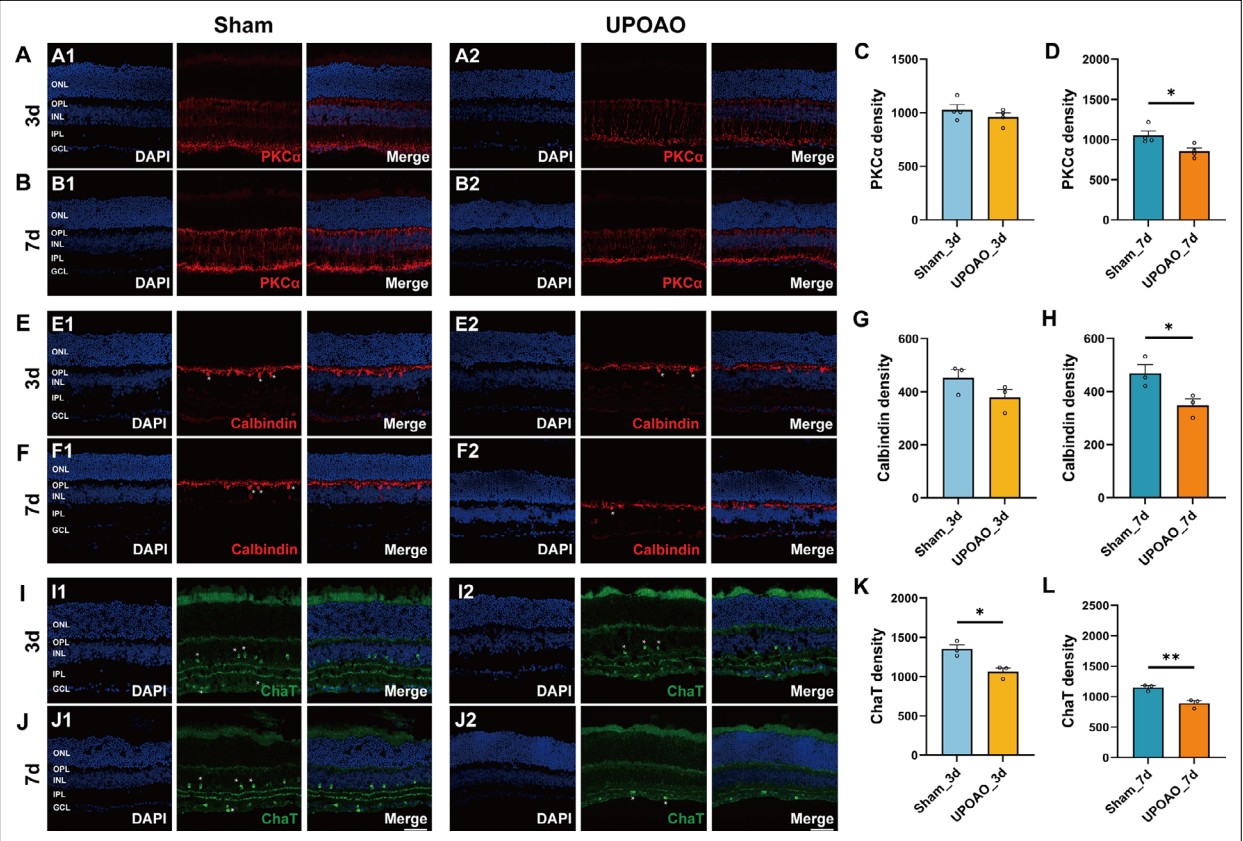

**Figure 5.** Changes in bipolar cells, horizontal cells, and cholinergic amacrine cells in unilateral pterygopalatine ophthalmic artery occlusion (UPOAO) mice. (**A**) Representative images of mouse retina co-stained with DAPI and PKCα at 3 days (**B**) Representative images of mouse retina co-stained with DAPI and PKCα at 7 days. (**C**) Quantification of PKCα fluorescence density at 3 days. (**D**) Quantification of PKCα fluorescence density at 7 days. n=4. (**E**) Representative images of mouse retina co-stained with DAPI and Calbindin at 3 days. (**F**) Representative images of mouse retina co-stained with DAPI and Calbindin at 7 days. Horizontal cell somata are indicated by white asterisks. (**G**) Quantification of Calbindin fluorescence density at 3 days. (**H**) Quantification of Calbindin fluorescence density at 7 days. n=3. (**I**) Representative images of mouse retina co-stained with DAPI and ChAT at 3 days. (**J**) Representative images of mouse retina co-stained with DAPI and ChAT at 7 days. Cholinergic amacrine cell somata are indicated by white asterisks. (**K**) Quantification of ChAT fluorescence density at 3 days. (**L**) Quantification of ChAT fluorescence density at 7 days. n=3. Data were presented as means ± SEM, *p<0.05, **p<0.01, ***p<0.001, t-test. Scale bar = 50 μm.

The online version of this article includes the following figure supplement(s) for figure 5:

**Figure supplement 1.** Changes in photoreceptor cells in unilateral pterygopalatine ophthalmic artery occlusion (UPOAO).

mitochondrial tricarboxylic acid cycle (*Figure 6—figure supplement 2C, D*). These results suggested that mitochondria and metabolism play significant roles in IRI.

In the 7 days reperfusion group, 429 genes were upregulated, and 310 genes were downregulated (*Figure 6G*). The DEGs at this stage were enriched in pathways related to immune effector processes, positive regulation of responses to external stimuli, and inflammation (*Figure 6H*). A heatmap was generated to show the upregulated and downregulated genes of immune inflammation-related pathways at 7 days after reperfusion in UPOAO, and most hub genes, such as *Cd86*, *Cd48*, *Tlr4*, and *Tlr6* were also enriched in these pathways (*Figure 6I*, *Figure 6—figure supplement 3A*). GSEA also showed similar results (*Figure 6—figure supplement 3B*). Moreover, RT–qPCR confirmed the upregulation of immune inflammation-related gene expression (*Figure 6—figure supplement 4*). Significantly, 112 genes that overlapped between immune-related genes and DEGs at 7 days post-UPOAO were primarily associated with immune receptor activity and phagocytic vesicles (*Figure 6—figure supplement 3C, D*). These results underscore the predominant role of immune responses during this stage.

To explore the associations between changes observed at 3 days and 7 days after reperfusion, we analyzed the DEGs and identified 17 overlapping genes (*Figure 6—figure supplement 5A*), which

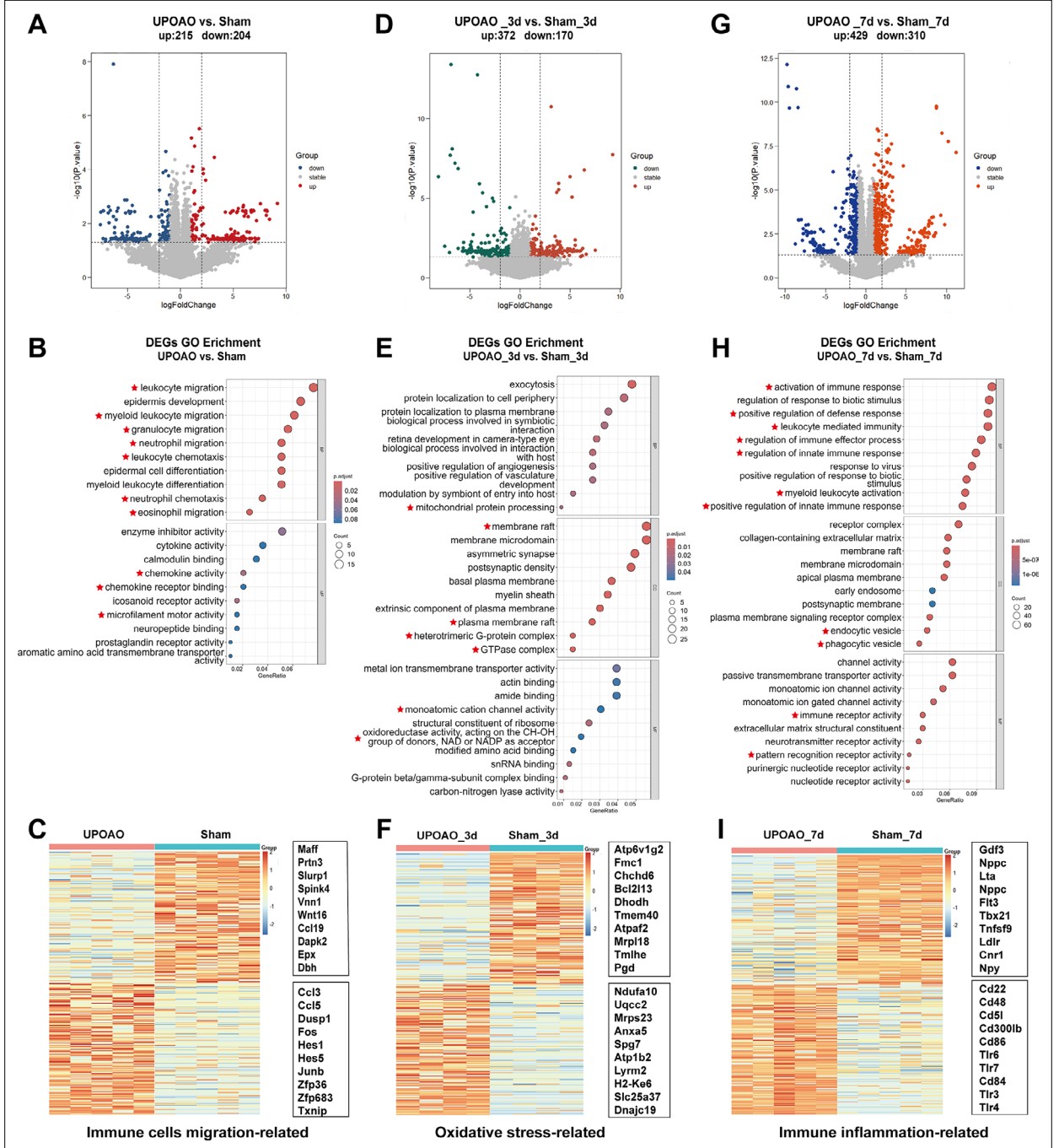

**Figure 6.** Transcriptomic features at different times of reperfusion. RNA-seq evaluation was performed at 0 day, 3 days, and 7 days reperfusion periods in unilateral pterygopalatine ophthalmic artery occlusion (UPOAO) revealing enrichment in pathways related to immune cell migration, oxidative stress, and immune inflammation. (**A, D, G**) Volcano plots display differential expression genes (DEGs) between UPOAO and sham eyes in the no-perfusion group (**A**), 3 days perfusion group (**D**), and 7 days perfusion group (**G**), respectively. Red dots: significantly upregulated genes, green and blue dots: significantly downregulated genes, and gray dots: stable expressed genes, adjusted p<0.05. log2FC = 1. (**B, E, H**) Gene ontology (GO) analysis of differential genes in the non-perfusion group (**B**), 3 days group (**E**), and 7 days group (**H**). The DEGs between UPOAO and sham eyes were enriched in pathways associated with immune cell migration (0d), oxidative stress (3d), and immune inflammation (7d), respectively. (**C, F, I**) Heatmap displaying the top 100 downregulated and top 100 upregulated DEGs between UPOAO and sham at non-perfusion, 3 days, and 7 days reperfusion, respectively. The box represents the genes related to immune cell migration (**C**), oxidative stress (**F**), and immune inflammation (**I**). The ranking was determined by the magnitude of fold change. In each heatmap, the upper box represents the top 10 downregulated genes, while the lower box represents the top 10 upregulated genes.

The online version of this article includes the following figure supplement(s) for figure 6:

*Figure 6 continued on next page*

*Figure 6 continued*

**Figure supplement 1.** Hub genes in protein-protein interaction (PPI) analysis and gene set enrichment analysis (GSEA) during the non-reperfusion stage in unilateral pterygopalatine ophthalmic artery occlusion (UPOAO).

**Figure supplement 2.** Hub genes, gene set enrichment analysis (GSEA) analysis at 3 days reperfusion in unilateral pterygopalatine ophthalmic artery occlusion (UPOAO), and relation with mitochondrial genes.

**Figure supplement 3.** Hub Genes, gene set enrichment analysis (GSEA) analysis at 7 days reperfusion in unilateral pterygopalatine ophthalmic artery occlusion (UPOAO), and relation with immune genes.

**Figure supplement 4.** Upregulation of immune inflammation-related gene expression in the 7 days reperfusion group.

**Figure supplement 5.** Co-expressed genes during 3 days and 7 days reperfusion.

were mainly associated with the nitric oxide biosynthetic process (*Figure 6—figure supplement 5B*). Our findings suggest that the immune inflammatory response observed after 7 days of reperfusion may represent the cumulative effect of the acute oxidative stress response during the initial 3 days reperfusion period.

## Leukocyte infiltration and microglial activation

Time course transcriptome analysis underscored the predominant role of immune inflammatory-responses during the reperfusion stage. Previous reports have indicated that the inflammatory response in retinal ischemia-reperfusion injury (RIRI) is orchestrated by peripheral immune cells and resident immune cells within the retina (*Li et al., 2018*; *Minhas et al., 2016*). In our study, we demonstrated the presence of leukocyte infiltration and microglial activation in the UPOAO model. Immunofluorescence staining for CD45 was performed on retinas at different time points post modeling to visualize the distribution and quantity of white blood cells (*Figure 7A–D*). Laser confocal Z-plane projections confirmed minimal leukocyte infiltration in the retinas of sham eyes, which showed that CD45+ cells in the vascular lumen likely represented patrolling cells (*Figure 7A*). A significant increase in CD45+ cells was observed in UPOAO model retinas after 1 day (*Figure 7E*), with a progressive increase in quantity over time (*Figure 7F and G*). Notably, the majority of CD45+ cells in UPOAO model retinas at 1 day displayed a morphology similar to that of vascular leukocytes (*Figure 7B*), suggesting a large influx of peripheral white blood cells into the retinal tissue. In contrast, CD45+ cells at 3 days and 7 days of reperfusion exhibited a combination of amoeboid and branched morphologies (*Figure 7C and D*), indicative of activated states.

Microglia, which are known to undergo activation and morphological changes in response to various insults, were examined in the superficial retinas of sham and UPOAO eyes. Immunofluorescence staining for Iba1 was conducted on flat-mounted retinas at 3 days and 7 days post-reperfusion. In sham eyes, Iba1+ microglia displayed small somas and elongated dendrites that were evenly distributed within the retina (*Figure 7H, I1*). Conversely, UPOAO retinas exhibited numerous activated Iba1+ microglia with enlarged somas, shortened dendrites, and an amoeboid appearance (*Figure 7H2, I2*). The number of Iba1+ microglia in UPOAO retinas was approximately five times greater than that in sham eyes at 3 days (*Figure 7J*), with a further increase observed at 7 days (*Figure 7K*), although this increase was less pronounced than that at the 3 days time point (*Figure 7J*). These results indicated a potential reduction in immune-mediated inflammation over time.

## RNA-seq comparison between UPOAO and extravascular occlusion models

To further study the characteristics of UPOAO, we analyzed the transcriptomes of two extravascular occlusion models: HIOP and UCCAO. The HIOP model induces ischemia through anterior chamber perfusion with normal saline (*Figure 8A*), while the UCCAO model involves ligation of the unilateral common carotid artery (*Figure 8H*). RNA-seq analysis of retinas subjected to 60 min of ischemia followed by 7 days of reperfusion in the HIOP model revealed enrichment of immune-related pathways, specifically highlighting increased activity in leukocyte-mediated immunity and regulation of immune effector processes (*Figure 8B*). This finding was consistent with the results of GO analysis, which revealed associations of the overlapping DEGs between the HIOP and UPOAO models with toll-like receptor binding, T-cell receptor binding, and immune-related functions (*Figure 8C and D*). To further explore these DEGs, we explored the remaining DEGs in the HIOP model in addition

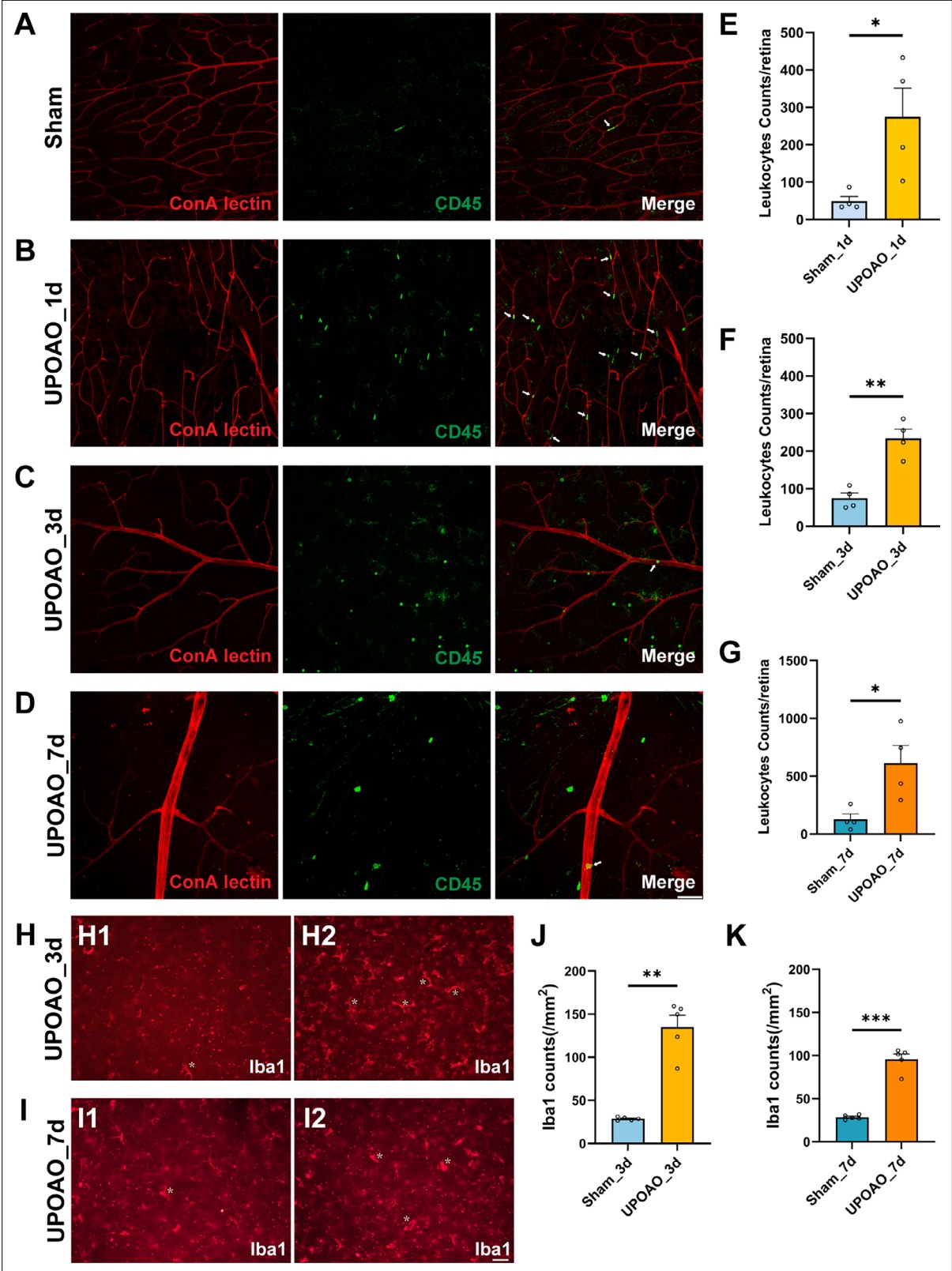

**Figure 7.** Peripheral leukocyte infiltration and retinal resident microglial activation. Rhodamine-labeled canavalin A was used for immediate cardiac perfusion to visualize blood vessels, followed by CD45 immunofluorescent staining to observe the relationship between blood vessels and CD45+ cells in sham (**A**), 1 day perfusion group (**B**), 3 days perfusion group (**C**), and 7 days perfusion group (**D**). The presence of CD45+ cells within blood vessels is indicated by white arrows. Scale bar = 50 μm. CD45+ cell counts were performed in whole retinas of 1 day (**E**), 3 days (**F**), 7 days (**G**). The cellular

*Figure 7 continued on next page*

*Figure 7 continued*

morphology and distribution of microglial cells in the superficial retina were assessed in 3 days (**H**) and 7 days (**I**). Activated microglial cells are indicated by white asterisks. Microglial cell counting was conducted in the superficial retina for 3 days (**J**) and 7 days (**K**). Data points for CD45$^+$ cells were derived from four flat-mounted retinas, and data points for microglial cells were from five flat-mounted retinas. Data were presented as means ± SEM, *p<0.05, **p<0.01, ***p<0.001, ****p<0.0001, t-test. Scale bar = 50 μm.

to the overlapping genes. GO analysis revealed that the remaining DEGs in the HIOP model were also enriched in immune responses, such as the T-cell receptor complex pathway, adaptive immune response, and B-cell-mediated immunity function (*Figure 8E*). Remarkably, upon examining the remaining DEGs specific to the UPOAO model, GO analysis revealed DEGs distinctly related to lipid and steroid metabolic processes (*Figure 8F and G*). This observation indicates that the UPOAO model, which is closely related to RAO, involves not only conventional immune responses but also unique regulation of lipid and steroid metabolism.

Similarly, the UCCAO model exhibited associations with negative regulation of B-cell proliferation, lymphocyte activation, and related processes (*Figure 8I*). Additionally, the hub genes identified through PPI analysis are presented (*Figure 8J*). However, the UCCAO model displayed minimal overlap with the UPOAO model, with only 15 overlapping DEGs identified in the Venn diagram analysis (*Figure 8K*). Further analysis of the UCCAO DEGs revealed two genes (*Mettl14* and *Ythdf2*) related to m6A modifications (*Figure 8L*). Although arterial blood flow is occluded in both the UPOAO and UCCAO models, our results revealed that the pathophysiological processes of the two models were particularly different after 7 days of reperfusion.

## Discussion

This study aimed to develop a more appropriate mouse model for investigating the ischemia-reperfusion process in RAO and unravelling the underlying pathophysiological mechanisms. We combined silicone wire embolization with carotid artery ligation to effectively block the blood supply to the retinal artery, closely mimicking the characteristics of the acute interruption of blood supply in RAO patients (*Biousse et al., 2018*). In the UPOAO model, a 60 min ischemia period was sufficient to simulate the injury of major retinal neural cells, such as RGCs, BCs, HCs, and cholinergic amacrine cells, resulting in visual impairment. Moreover, histologic examination demonstrated thinning of the inner layer of the retina, especially the GCL. Time course transcriptome analysis revealed various pathophysiological processes related to immune cell migration, oxidative stress, and immune inflammation during non-reperfusion and reperfusion periods. The resident microglia within the retina and peripheral leukocytes that access the retina were markedly increased during reperfusion periods. Additionally, we compared the transcriptomic signatures of the UPOAO model with those of two commonly used ischemia-reperfusion mouse models (HIOP and UCCAO), highlighting the potential relevance of the UPOAO model to ocular vascular occlusive diseases.

The UPOAO mouse model effectively simulates the characteristic features of RAO observed in patients. RAO patients often exhibit distinctive patterns in dark-adapted ERGs, where b-waves are reduced while a-waves remain stable (*Block and Schwarz, 1998*; *Lima et al., 2010*; *Suzuki et al., 2014*). In our study, we observed almost the same phenomenon in ERG amplitude after 60 min of ischemia in the UPOAO model (*Supplementary file 1*). This observation was further confirmed by immunofluorescence staining of bipolar cells and photoreceptor cells. Importantly, a previous study linked the thinning of the inner retina with alterations in ERG in patients with RAO (*Shinoda et al., 2008*). Our results showed that ischemic damage predominantly affected the inner retinal layers, while the outer layer was almost unaffected (*Supplementary file 1*). This distinction is attributed to the reliance of the inner layers on the retinal artery for blood supply, whereas the outer layers are supplied by the choroid. Additionally, histological examination revealed increased retinal NFL and GCL thickness and decreased IPL thickness after 3 days of reperfusion, indicating that ischemia-induced oedema had not completely subsided.

This study comprehensively explored the transcriptomic signature after ischemia and revealed that resident and peripheral immune cells may play major roles in pathological processes. In the non-reperfusion group, the transcriptional changes were primarily involved in the migration of immune cells such as leukocytes and neutrophils from blood vessels into ischemic retinal tissue. This process

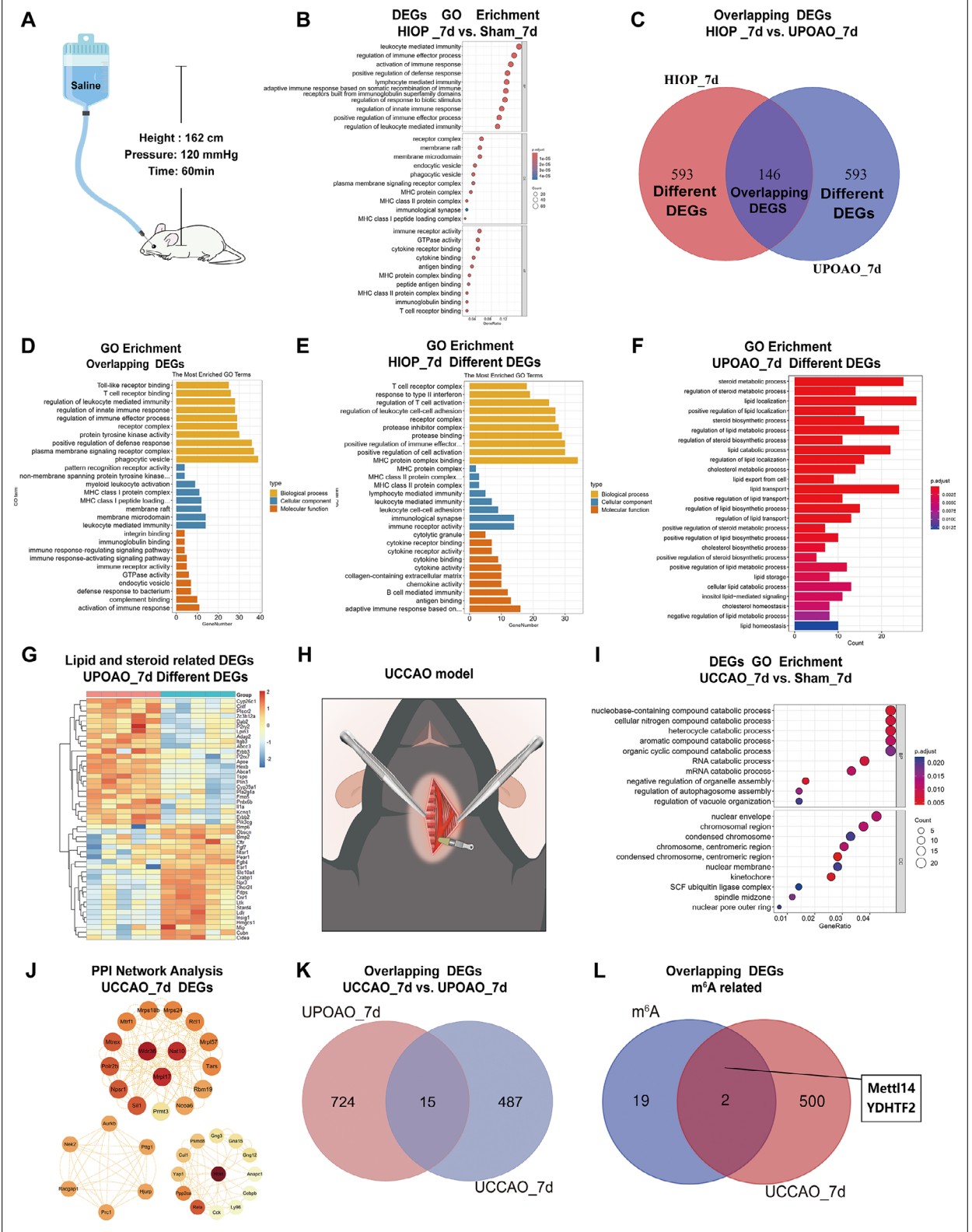

**Figure 8.** Transcriptomic results comparison between unilateral pterygopalatine ophthalmic artery occlusion (UPOAO), high intraocular pressure (HIOP), and unilateral common carotid artery occlusion (UCCAO) models. RNA-seq comparison between UPOAO and extravascular occlusion models: HIOP and UCCAO. (**A**) Schematic illustration of the HIOP model. (**B**) Gene ontology (GO) analysis of differential expression genes (DEGs) in the high intraocular pressure (HIOP) model at 7 days perfusion. (**C**) Venn diagram indicating the overlapping DEGs (146 genes) between HIOP and UPOAO

*Figure 8 continued on next page*

*Figure 8 continued*

models, as well as the remaining DEGs specific to each group (593 genes in each group). (**D**) Gene ontology (GO) analysis of the overlapping DEGs between HIOP and UPOAO models. (**E**) GO analysis of the remaining DEGs in the HIOP model at 7 days perfusion, excluding the overlapping DEGs. (**F**) GO analysis of the remaining DEGs, excluding the overlapping DEGs. (**G**) Heatmap showing the inter-sample distribution of lipid and steroid-related DEGs from the analysis of the remaining DEGs in the UPOAO model. (**H**) Schematic illustration of the UCCAO model. (**I**) GO analysis of DEGs in the UCCAO model. (**J**) Hub genes were identified through protein-protein interaction (PPI) network analysis of the DEGs in the UCCAO model. (**K**) Venn diagram indicating the overlapping DEGs between UCCAO and UPOAO models (15 genes). (**L**) Venn diagram indicating the presence of two overlapping DEGs between UCCAO DEGs and m6A-related genes.

is closely linked to injury during ischemia and reperfusion, where leukocytes and neutrophils infiltrate neural tissue through the vascular endothelium (*Szabo et al., 1991*; *Szabo, 1992*). Our results revealed an increase in CD45[+] leukocytes in the retina and retinal microvasculature, suggesting that these leukocytes were recruited from the bloodstream to the damaged retinal tissue following reperfusion. During leukocyte access to the retina, chemokines, chemokine receptors, adhesion molecules, and cytoskeletal components play important roles in regulating endothelial permeability and facilitating leukocyte adhesion to endothelial cells (*Johnson-Léger et al., 2000*; *Springer, 1994*). Our findings of related gene expression are consistent with this mechanism.

During the 3 days reperfusion period, we observed that an increased number of DEGs were significantly involved in the critical pathological response to oxidative stress in the context of IR tissues. The generation of reactive oxygen species (ROS) in mitochondria and subsequent oxidative stress are widely recognized as major causes of retinal cell damage induced by IR (*Fang et al., 2015*; *Oharazawa et al., 2010*). This finding suggested that mitochondrial function, which is involved in oxidative stress processes, may play an important role in the pathophysiology of IRI during the early reperfusion stage. Oxidative stress can influence the permeability of the inner blood-retina barrier (iBRB), allowing leukocytes to access retinal tissues (*Kaur et al., 2008*). The infiltration and activation of immune cells are recognized as the underlying causes of many retinal diseases, including ischemic ophthalmia (*Kaur et al., 2008*; *Eltzschig and Collard, 2004*). In the UPOAO model, the increase in CD45[+] leukocytes and Iba1[+] microglia in the retina after 3 days of reperfusion supports this viewpoint.

In the later stage of reperfusion (7 days), the DEGs were mainly enriched in immune regulation and inflammation. With more upregulated DEGs than downregulated ones, we hypothesize that the activation of the immune inflammatory response contributes to further IRI in retinal tissue. Previous studies have shown that the retina can elicit immunological responses during ischemia-reperfusion injury and that immune inflammation is an important phenomenon in the progression of this injury (*Minhas et al., 2016*). The excessive ROS generated by mitochondria in RGCs during the activation of inflammatory responses can damage cell structure and visual function (*Sanderson et al., 2013*). Microglia, considered the primary resident immune cells, contribute to inflammatory responses and consequent neural damage (*Abcouwer et al., 2021*). Infiltrating leukocytes can activate resident microglia, forming a feedback loop that exacerbates inflammation (*Au and Ma, 2022*). Complex interactions between immune cells and retinal neurons after RIRI have been reported (*Qin et al., 2022*). Interestingly, in our UPOAO model, the death of major retinal neural cells, such as RGCs, BCs, and HCs, was correlated with an increase in infiltrating leukocytes and resident microglia. Therefore, we considered that the immune response and neuroinflammation observed after 7 days of reperfusion may result from the cumulative effects of acute oxidative stress and the responses of resident and peripheral immune cells (*Figure 9*, *Supplementary file 1*). In summary, we described the pathological processes at different time points and highlighted the important role of resident and peripheral immune cell responses in the UPOAO model. This finding offers valuable insights for subsequent screening of pathogenic genes and potential immunotherapy approaches.

Compared with the HIOP and UCCAO models, the UPOAO model is a more appropriate choice for studying retinal IRI in RAO. The HIOP model, widely used for investigating the pathogenesis of APACG, exhibits retinal degeneration features similar to those of the UPOAO model during both reperfusion periods (*Yao et al., 2023*). In our UPOAO model, histologic and immunohistochemical analysis demonstrated that 60 min of ischemia followed by 3 days or 7 days of reperfusion can cause irreversible damage to the retinal structure and visual function. Specifically, we observed distinct alterations, such as thinning of the inner retina, substantial RGCs apoptosis, and decreased b-wave amplitude. Multiple studies on the HIOP model have reported consistent results (*Abcouwer et al.,*

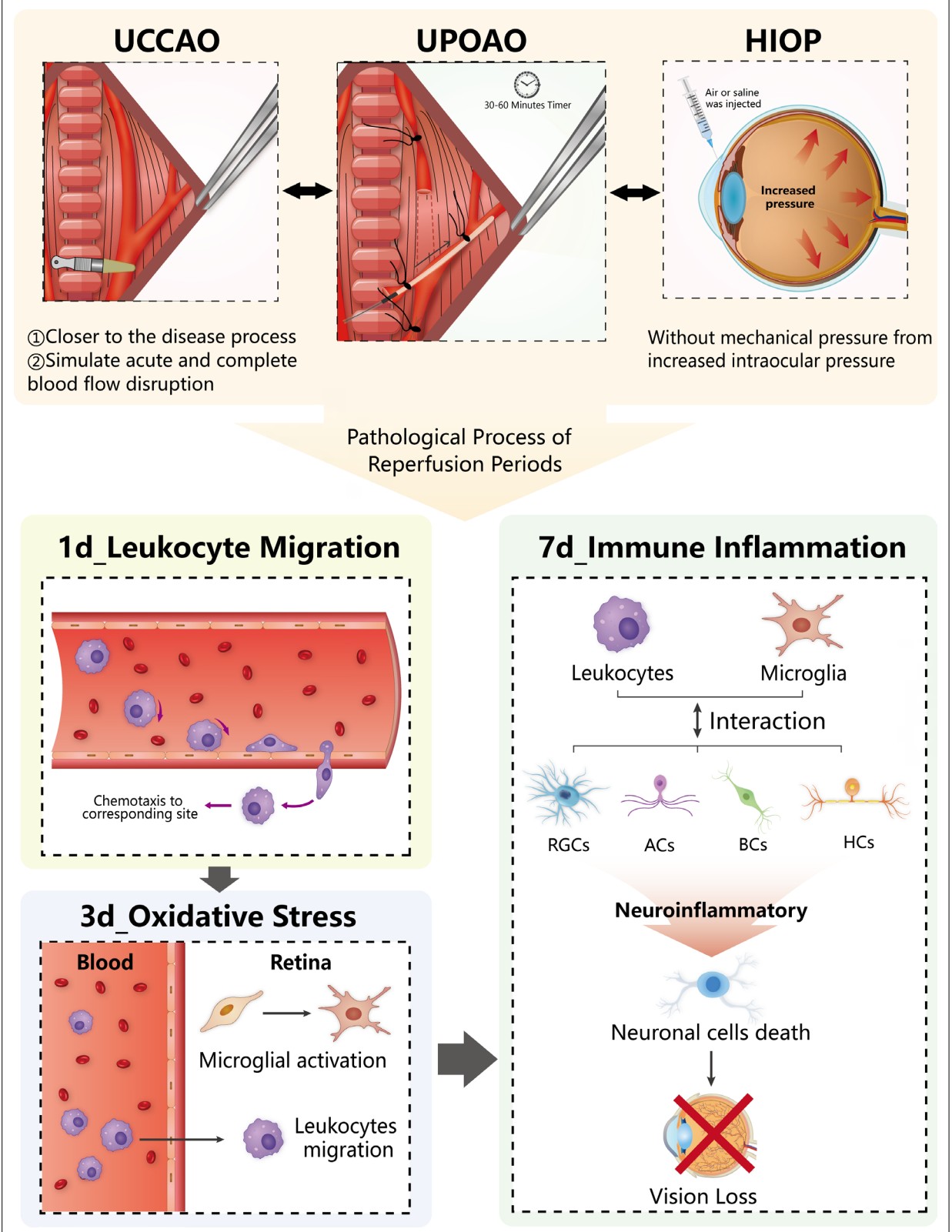

**Figure 9.** The characteristic features of unilateral pterygopalatine ophthalmic artery occlusion (UPOAO) model during ischemia-reperfusion periods.

*2021*; *Zeng et al., 2023*; *Yang et al., 2020*). However, some typical phenotypes in the HIOP model, such as thinning of the outer nuclear layer and the entire neuroretina and decreased a-wave amplitude, were not observed in the UPOAO model. Moreover, our transcriptome sequencing revealed a specific subset of DEGs unique to the UPOAO model, including *Apoe*, *Abca1*, *Ldlr*, *Cyp39a1*, and *Bmp6*. Notably, these DEGs were significantly enriched in pathways associated with lipid and steroid synthesis. Steroid and lipid metabolism homeostasis is disrupted under conditions of oxidative stress and inflammation (*Lam et al., 2016*). In the UPOAO model, the observed lipid and steroid biological processes may result from the cumulative pathological responses triggered by IRI, as observed in the HIOP model. In the UCCAO model, RNA-seq revealed that DEGs were enriched in two main pathways: (1) epigenetic modification-related pathways, including nucleobase-containing compound catabolic process, RNA catabolic process, mRNA catabolic process and N6-methyladenosine (m6A) modification; and (2) cell death pathways, including regulation of autophagosome assembly, negative regulation of neuron death, and negative regulation of the neuronal apoptotic process. Epigenetic mechanisms may play a key role in the pathophysiology of ocular disease (*He et al., 2013*). m6A, one of the most common RNA modifications, has been reported to regulate cell death processes, including apoptosis and autophagy, in the pathological process of IRI (*Wang et al., 2022*). These results suggest that epigenetic mechanisms may significantly influence cell death during the 7 days reperfusion period in the UCCAO model. Lee et al. evaluated the characteristics of UCCAO without reperfusion using visual, histological, and immunohistochemical approaches. Their results revealed delayed perfusion of the ipsilateral retina, thinning of the inner retinal layer 10 weeks after surgery, and a dramatic decrease in the amplitudes of b-waves on day 14 after UCCAO (*Lee et al., 2020*; *Lee et al., 2021*). This finding suggested that UCCAO primarily represents a model of retinal hypoperfusion injury and may not effectively reflect acute ischemic-induced structural and functional damage in RAO (*Figure 9*). These results led us to consider the UPOAO model an effective experimental model for studying the pathological processes underlying acute ischemia and IRI.

Our research has certain limitations that should be acknowledged. First, our study focused on the changes that occurred within 60 min of ischemia and within the first 7 days of reperfusion in the UPOAO model. Further exploration is needed to understand the changes induced by longer reperfusion periods. Additionally, we proposed possible pathological mechanisms, including iBRB damage, oxidative stress, and immune inflammation, mainly based on the enrichment results of DEGs at three reperfusion time points. More molecular experimental validation is required to substantiate these proposed mechanisms.

In summary, by thoroughly exploring the injury to major retinal neural cells, visual impairment, and pathophysiological changes in the UPOAO model, we confirmed that this model can effectively simulate the acute ischemia-reperfusion processes in RAO. It serves as an ideal mouse model for investigating the underlying pathological mechanisms of ischemia and reperfusion. Furthermore, our UPOAO model holds great promise as a novel model for studies on pathogenic genes and potential therapeutic interventions for RAO.

## Materials and methods

### Animals

Eight-week-old male C57BL/6 mice were used in the experiments. Only male mice were used to exclude the potential influence of oestrogenic hormones. The mice were provided sufficient food and water and were maintained on a 12 hr dark-light cycle in a room with regulated temperature conditions. All the experimental procedures were designed and conducted according to the ethical guidelines outlined in the Association for Research in Vision and Ophthalmology (ARVO) Statement for the Use of Animals in Ophthalmic and Vision Research. The study protocol and methods received approval from the Experimental Animal Ethics Committee of Renmin Hospital of Wuhan University (approval number: WDRM-20220305A).

### Preparation and surgical procedure of the UPOAO model

Eight-week-old male C57BL/6 mice weighing 20–25 grams were used in this study. An isoflurane-based anaesthesia system was used to induce and maintain general anaesthesia in the mice during surgical procedures. The mice were anaesthetized with a 1.5–2% concentration of isoflurane delivered in a mixture of

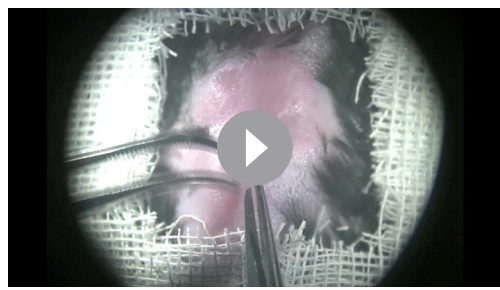

**Video 1.** Blunt dissection of cervical vessels. The neck was exposed, and the cervical arteries were separated using blunt dissection techniques.

https://elifesciences.org/articles/98949/figures#video1

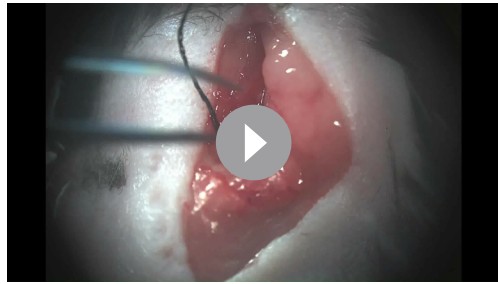

**Video 2.** Insertion of silicone wire embolus and induction of ischemia. The silicone wire embolus was inserted into the PPA, initiating the ischemic condition. PPA: the pterygopalatine artery.

https://elifesciences.org/articles/98949/figures#video2

nitrous oxide and oxygen via rubber tubing. Body temperature was maintained at 37±0.5°C throughout the surgery. Surgical instruments were sterilized with 75% ethanol before use to ensure sterility.

We drew inspiration from the widely employed middle cerebral artery occlusion (MCAO) model (*Chiang et al., 2011*; *Sasaki et al., 2009*), commonly used in cerebral ischemic injury research, which guided the development of the UPOAO model. The mouse was gently positioned in a supine posture on a heating blanket, and its neck was exposed. The preparation involved depilation of the neck area, followed by skin disinfection, and finally, a midline incision along the neck was made. After separating the neck gland using two tweezers, careful blunt dissection was employed to separate the left CCA, internal carotid artery (ICA), and external carotid artery (ECA), avoiding the compression of nearby nerves and veins (*Figure 1A*) (*Video 1*). Then, the CCA and ICA were ligated with a 6–0 suture. A knot was secured using a 6–0 suture at the distal end of the ECA, and a slipknot was created at the proximal end (*Figure 1B*). To preserve the reperfusion process and avoid bleeding and mortality during surgery, a silicone wire embolus was inserted through the ECA instead of through the CCA. Ophthalmic scissors were used to make a small inverted 'V'-shaped incision between the two suture knots on the ECA. A specialized silicone wire embolus, measuring 7±0.1 mm in length and 0.21±0.1 mm in diameter, was inserted through the incision, directing it into the ECA and further into the CCA (*Figure 1C*). Next, the ligation on the ICA was removed, and the ECA was cut at the point where the artery had been previously incised with scissors. Subsequently, the silicone wire embolus was retracted to the bifurcation of the CCA, rotated counterclockwise, and inserted into the ICA and further into the pterygopalatine artery (PPA) (*Figure 1D1*). The silicone tail of the silicone wire embolus was positioned near the bifurcation of the CCA (*Figure 1D2*), effectively obstructing the ophthalmic artery (OA) (*Video 2*). Then, the slipknot was fastened, and the skin was sutured. The mouse was free to move around during arterial embolization.

Following a predetermined ischemia embolization time, the silicone wire embolus was carefully removed from the PPA without massive haemorrhage (*Figure 1E*). The CCA suture was removed, restoring arterial reperfusion (*Figure 1F*) (*Video 3*). Then, the skin wound was sutured, and the mice were routinely fed during reperfusion (*Figure 1G*).

## Staining of retinal vessels

To clearly visualize the retinal vessels, rhodamine-labeled canavalin A was used for mouse heart perfusion. Each mouse was anaesthetized with 1% sodium pentobarbital and placed in the supine position on a foam board resting on a tray. The sternum was opened to expose the heart. A needle attached to the perfusion tube was inserted into the left ventricle via the apex of

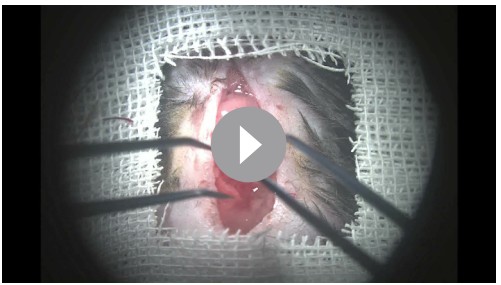

**Video 3.** Extraction of silicone wire embolus and reperfusion. The silicone wire embolus was removed, allowing reperfusion to occur. The mouse's wound was sutured, and standard feeding procedures were followed.

https://elifesciences.org/articles/98949/figures#video3

the mouse heart and secured. The right atrial appendage was dissected with scissors to allow blood outflow. Saline solution was perfused through the left ventricle using a pump system to drain blood from the arteries. Subsequently, the prepared solution of rhodamine-labeled canavalin A was injected into the mice and circulated throughout the mouse vessels for 30 min. The mice were then sacrificed, and their eyes were removed and immersed in paraformaldehyde (PFA) for 1 hr in the dark. A flat-mounted retina was obtained, and retinal images were captured using a Leica SP8 confocal microscope equipped with a 10×objective.

## Quantification of RGCs and microglia

The mice were euthanized by cervical dislocation, and their eyes were fixed in 4% PFA for 60 min at room temperature (RT). Subsequently, the cornea and lens were excised, and the intact retina was isolated for retinal flat mounting. The retina was immersed in PBS supplemented with 5% bovine serum albumin (BSA) and 0.5% Triton X-100 for overnight blocking and then incubated with Brn3a antibody (Synaptic Systems, Germany) or Iba1 antibody (Wako, Japan). Following a two-day incubation with the primary antibody, the retina was gently washed three times with PBS. Subsequently, the retina was incubated with Alexa Fluor 594 (AntGene, Wuhan, China) in a light-protected cassette for two days. After three rinses, the retina was uniformly sectioned into a four-leaf clover morphology and flattened using a coverslip. Retinal flat mounts labeled with Brn3a were photographed to count using a fluorescence microscope (BX51, Olympus, Japan) and representative pictures were imaged using a Leica SP8 confocal microscope (Leica TCS SP8, Germany). Retinas labeled with Iba1 were imaged utilizing a fluorescence microscope (BX63; Olympus, Tokyo, Japan). Each quadrant of the retina was systematically subdivided into central, middle, and peripheral fields (distance from the optic nerve head: central field: 0.1 mm to 0.5 mm, middle field: 0.9 mm to 1.3 mm, and peripheral field: 1.7 mm to 2.1 mm). The surviving RGCs and activated microglia of 12 fields in each retina were quantified and averaged using ImageJ software (National Institutes of Health, USA).

## Electroretinogram (ERG)

Mice were dark-adapted overnight before ERG, and all subsequent procedures were carried out in the dark. Before ERG, the mice were anaesthetized with 1% sodium pentobarbital via intraperitoneal injection, and their pupils were dilated. A subcutaneous electrode was inserted into the posterior cervical skin, a tail electrode was affixed to the posterior end of the mouse tail, and the corneal electrode was gently placed on the central corneal surface. A RetiMINER-C visual electrophysiological system (3VMED Co., Ltd., Shanghai, China) was used for recording electrical responses. The a-waves, b-waves, and oscillatory potentials (OPs) after flash stimuli of 0.01, 0.03, 0.1, 0.3, 1.0, 3.0, and 10.0 cd.s/m *Hayreh and Zimmerman, 2005* in scotopic adaptation were recorded. The amplitudes and implicit times of a-waves, b-waves, and OPs in response to various flash stimuli were analyzed.

## Optical coherence tomography imaging (OCT) and fluorescein fundus angiography (FFA)

The mice were placed on the platform of a Spectralis HRA + OCT device (Heidelberg Engineering, Heidelberg, Germany) for OCT imaging following anaesthesia. Pupil dilation was performed, and normal saline was applied regularly to maintain corneal moisture. The focal length of the device was adjusted until the mouse retina was clearly visible. The head of the mouse was gently repositioned to capture images of the peripheral fundus. The multiline mode was used to scan each layer of the retina, and four quadrants of view centred on the nipple in the upper left, lower left, upper right, and lower right were recorded. In this study, the total retinal thickness was manually segmented into three parts and encompassed the entire thickness from the nerve fibre layer to the photoreceptor layer. The ganglion cell complex (GCC) was defined as the combined thickness of the retinal nerve fibre layer (RNFL), ganglion cell layer (GCL), and inner plexiform layer (IPL). The inner nuclear layer (INL) and the outer plexiform layer (OPL) were combined for thickness analysis due to the difficulty in distinguishing between these layers. The remaining retinal layers, including the outer nuclear layer (ONL), inner segment/outer segment (IS/OS), and retinal pigment epithelium (RPE) layers, were measured together. The thickness of the retina at 1.5 papillary diameters (PD), 3.0 PD, and 4.5 PD from the centre of the optic disc was measured using the Heidelberg measuring tool.

**Table 1.** Antibodies used in staining of flat-mounted retina and sections.

| Antibodies | Species | Dilution | Company |
|---|---|---|---|
| PKCα | rabbit | 1:1000 | Sigma |
| Recoverin | rabbit | 1:1000 | Millipore |
| Calbindin | rabbit | 1:5000 | SWANT |
| ChAT | goat | 1:200 | Millipore |
| Iba1 | rabbit | 1:500 | Wako |
| CD45 | rabbit | 1:100 | CST |
| Rhodamine-labeled canavalin A | / | 1:125 | MKBio |
| Goat anti-mouse IgG Alexa Fluor 488 | goat | 1:500 | Jackson |
| Donkey anti-rabbit IgG Alexa Fluor 594 | donkey | 1:500 | Jackson |
| Donkey anti-goat IgG Alexa Fluor 488 | donkey | 1:500 | Jackson |

For FFA, the mice were anaesthetized, and their pupils were dilated. Subsequently, fundus angiography was performed immediately following an intraperitoneal injection of sodium fluorescein. Images were acquired alternately for both eyes within 5 s of the injection.

## Hematoxylin and eosin (HE)

Mouse eyes were fixed in FAS eyeball fixation solution (Service-bio, Wuhan, China) for 48 hr, followed by dehydration and subsequent embedding in paraffin. The paraffin blocks were trimmed parallel to the optic nerve to obtain 3–4 μm thick sections where the optic nerve was located. Six paraffin sections from each eyeball were stained with HE. Images were captured using an Olympus fluorescence microscope. The retinal thickness within a range of 200–1100 μm from the optic disc was measured using Image-Pro Plus 6.0 software.

## Staining of retinal sections

For immunofluorescence, mouse eyes were fixed in 4% PFA solution and subjected to gradient dehydration using 10%, 20%, and 30% sucrose solutions. The following day, the eyes were embedded in optimum cutting temperature (OCT) compound (SAKURA, USA) and frozen at –80 °C. Several 14-μm-thick frozen sections were obtained from each eyeball through the use of a freezing microtome (Leica, Wetzlar, Germany). The section surface was parallel to the optic nerve, and 3–6 frozen sections were mounted on a single slide. The slides were stored at –20 °C until use.

All sections were blocked with 5% BSA and 0.5% Triton X-100 in PBS for 2 hr. Primary antibodies, as listed in *Table 1*, were diluted in PBS containing 5% BSA and 0.5% Triton X-100 and incubated at 4 °C overnight. The following day, after three rinses, the sections were incubated with secondary antibodies at RT for 2 hr in a cassette, followed by three washes. The sections were then stained with DAPI (Service-bio, Wuhan, China) for 15 min. After the final three rinses, the frozen sections were sealed. A Leica SP8 instrument equipped with a 40×objective was used to photograph frozen sections near the optic nerve head. The fluorescence intensity of all slices of each eye was determined using ImageJ software.

## Surgical procedure for the UCCAO model and the HIOP model

Animals for the UCCAO model were prepared and managed according to the procedure described for the UPOAO model. After blunt dissection, the left CCA was ligated with a 6–0 suture. After 60 min of ligation, the suture was removed, and the skin wound was subsequently closed with sutures. Mice were routinely fed during the reperfusion period, and the retinas of UCCAO mice were harvested 7 days post-surgery.

The HIOP model was generated based on previously established methods (*Chen et al., 2003*). Briefly, mice were anaesthetized by intraperitoneal injection of 1% sodium pentobarbital. Pupils were dilated using tropicamide phenylephrine eye drops (0.5% tropicamide and 0.5% deoxyadrenaline hydrochloride, Santen Pharmaceutical) applied 5 min in advance. An insulin hypodermic needle

attached to a silicone elastomer tube was inserted into the anterior chamber of the unilateral experimental eye. The IOP was elevated for 60 min by raising the height of the saline solution storage bag to 162 cm, after which the needle was removed to immediately restore the IOP to baseline levels. Iris whitening and loss of retinal red-light reflexes indicated retinal ischemia. After surgery, levofloxacin hydrochloride ophthalmic gel was applied to the eyes of the mice. Retinas from HIOP mice were collected 7 days post-surgery.

## Transcriptome sequence and analysis

Retinal samples from UPOAO, HIOP, and UCCAO mice were extracted and promptly frozen in liquid nitrogen within 10 min of cervical dislocation. Total RNA extraction was performed according to the protocol outlined in the TRIzol Reagent manual (Life Technologies, CA, USA). RNA integrity and concentration were assessed using an Agilent 2,100 Bioanalyzer (Agilent Technologies, Inc, Santa Clara, CA, USA). The resulting RNA samples were then pooled and subjected to sequencing on the Illumina HiSeq3000 platform in a 150 bp paired-end read format. The raw RNA-sequencing (RNA-seq) reads were preprocessed and quantified using the featureCounts function in the SubReads package version 1.5.3, with default parameters.

RNA-seq analysis was conducted on samples from five mice in the non-perfusion group, four mice in the 3 days perfusion group, and five mice in the 7 days perfusion group of the UPOAO model. Additionally, retinas from both the 7 days HIOP experimental eyes and bilateral UCCAO eyes were collected (n=5) for analysis.

Data normalization and subsequent processing utilized the 'limma' package in R software (version 4.2.3; *Ritchie et al., 2015*). Volcano plots and Venn diagrams were generated for visualization to identify significantly DEGs and perform Gene Ontology (GO) annotation analysis. DEGs with a |log2-fold change (FC)|>1 and adjusted p-value <0.05 were significantly differentially expressed. Enrichment analysis focused on DEGs with an adjusted p<0.05 and an enriched gene count >5. To visualize the expression patterns of significant DEGs, a heatmap was generated using the 'pheatmap' package. Gene Set Enrichment Analysis (GSEA) was conducted using the GSEA program (version 4.3.2) *Subramanian et al., 2005*, employing the default gene set m-subset (mh.all.v2023.1.mm.symbols.gmt) to explore significant functional and pathway differences. Enriched pathways were classified based on criteria such as adjusted p-value (<0.05), false discovery rate (FDR) q value (<0.25), and normalized enrichment score (|NES|>1). Furthermore, (PPI) analysis was performed using the Search Tool for the Retrieval of Interacting Genes (STRING) database (https://string-db.org/; *Szklarczyk et al., 2019*). Genes with a greater degree of protein-level interactions with others were further analyzed using Cytoscape software (version 3.8.2) to generate a downstream PPI map. PPI pairings with an interaction score >0.7 were extracted and visualized using Cytoscape 3.9.0 (*Kohl et al., 2011*).

## Real-time quantitative polymerase chain reaction (RT-qPCR)

RNA was extracted from fresh mouse retinal samples and subjected to reverse transcription using HiScript lll RT SuperMix for qPCR (Vazyme, China) following the manufacturer's instructions. RT–qPCR analysis was conducted on a CFX instrument (Bio-Rad) using AceQ qPCR SYBR Green Master Mix (Vazyme, China). The PCR program comprised 40 cycles of 10 s at 95 °C and 30 s at 60 °C. Assays were performed in triplicate, and Ct values were normalized to *Actb* levels. Relative quantification of target gene expression was calculated using the 2-ΔΔCt method. The primer sequences are provided in *Supplementary file 2*.

## Statistical analysis

All the data were analyzed using GraphPad Prism version 9.0 (GraphPad Software, San Diego, CA, USA). Each experimental group included a minimum of three biological replicates. T-tests, including RGCs and microglia counting, OPs analysis, retinal thickness measurements via HE staining, and fluorescence intensity quantification, were used to compare the two groups. Two-way ANOVA (two-way analysis of variance) test was used for more than two different experimental groups, including ERG waves and retinal thickness measurements in OCT. All data are presented as the mean ± standard error of the mean (SEM), and the statistical graphs are shown as scatter bar graphs and line charts. p<0.05 was considered to indicate statistical significance.

## Acknowledgements

The authors would like to express their gratitude to all participants involved in the experiments. The authors also wish to acknowledge the Eye Institute of Renmin Hospital of Wuhan University for providing us with the experimental facilities and environment. Special thanks are extended to Professor Shenqi Zhang for the technical support in the surgical procedure of the UPOAO model.

## Additional information

### Funding

| Funder | Grant reference number | Author |
|---|---|---|
| National Nature Science Foundation of China | 82371079 | Xuan Xiao |
| National Key R&D Program of China | 2023YFC2308404 | Xuan Xiao |
| Fundamental Research Funds for the Central Universities | 2042023gf0013 | Xuan Xiao |
| Key research and development project of Hubei Province | 2022BCA009 | Xuan Xiao |

The funders had no role in study design, data collection and interpretation, or the decision to submit the work for publication.

### Author contributions

Yuedan Wang, Ying Li, Conceptualization, Data curation, Validation, Methodology, Writing - original draft, Writing - review and editing; Jiaqing Feng, Formal analysis, Validation, Visualization, Writing - original draft, Writing - review and editing; Chuansen Wang, Validation, Methodology, Writing - review and editing; Yuwei Wan, Data curation, Validation; Bingyang Lv, Validation, Video production and editing; Yinming Li, Hao Xie, Ting Chen, Faxi Wang, Ziyue Li, Validation, Writing - review and editing; Anhuai Yang, Resources, Validation, Project administration; Xuan Xiao, Conceptualization, Resources, Funding acquisition, Validation, Project administration

### Author ORCIDs

Jiaqing Feng ⓘ https://orcid.org/0000-0003-0514-4663
Ziyue Li ⓘ https://orcid.org/0000-0002-7393-1442
Xuan Xiao ⓘ https://orcid.org/0000-0001-7762-8689

### Ethics

The study protocol and methods received approval from the Experimental Animal Ethics Committee of Renmin Hospital of Wuhan University (approval number: WDRM-20220305A).

Reviewer #1 (Public Review): https://doi.org/10.7554/eLife.98949.4.sa1
Reviewer #2 (Public Review): https://doi.org/10.7554/eLife.98949.4.sa2
Author response https://doi.org/10.7554/eLife.98949.4.sa3

## Additional files

### Supplementary files

• Supplementary file 1. **Summary table**. Time course of all the morphological, functional, cellular, and transcriptome changes in the UPOAO model.-: no change; ↑: increase; ↓: decrease; /: not involved in the article; #: supplementary content: also increase after 60 min-ischemia and 1d-reperfusion.

• Supplementary file 2. Primers used in this study.

- MDAR checklist

## Data availability

Sequencing data have been deposited in Dryad, Dataset URL: https://doi.org/10.5061/dryad.s1rn8pkhb.

The following previously published dataset was used:

| Author(s) | Year | Dataset title | Dataset URL | Database and Identifier |
|---|---|---|---|---|
| Wang Y, Li Y, Feng J, Wang C, Wan Y, Lv B, Li Y, Xie H, Chen F, Wang F, Li Z, Yang A, Xiao X | 2024 | Transcriptional responses in a mouse model of silicone wire embolization induced acute retinal artery Ischemia and Reperfusion | https://doi.org/10.5061/dryad.s1rn8pkhb | Dryad Digital Repository, 10.5061/dryad.s1rn8pkhb |

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
