## [Editor Report · eLife assessment]

The manuscript establishes a sophisticated mouse model for acute retinal artery occlusion (RAO) by combining unilateral pterygopalatine ophthalmic artery occlusion (UPOAO) with a silicone wire embolus and carotid artery ligation, generating ischemia-reperfusion injury upon removal of the embolus. This clinically relevant model is **useful** for studying the cellular and molecular mechanisms of RAO. The data overall are **solid**, presenting a novel tool for screening pathogenic genes and promoting further therapeutic research in RAO.

---

## [Referee Report · Reviewer #1 (Public Review)]

Summary:

Wang, Y. et al. used a silicone wire embolus to definitively and acutely clot the pterygopalatine ophthalmic artery in addition to carotid artery ligation to completely block blood supply to the mouse inner retina, which mimic clinical acute retinal artery occlusion. A detailed characterization of this mouse model determined the time course of inner retina degeneration and associated functional deficits, which closely mimic human patients. Whole retina transcriptome profiling and comparison revealed distinct features associated with ischemia, reperfusion, and different model mechanisms. Interestingly and importantly, this team found a sequential event including reperfusion-induced leukocyte infiltration from blood vessels, residual microglial activation, and neuroinflammation that may lead to neuronal cell death.

Strengths:

Clear demonstration of the surgery procedure with informative illustrations, images, and superb surgical videos.

Two time points of ischemia and reperfusion were studied with convincing histological and in vivo data to demonstrate the time course of various changes in retinal neuronal cell survivals, ERG functions, and inner/outer retina thickness.

The transcriptome comparison among different retinal artery occlusion models provides informative evidence to differentiate these models.

The potential applications of the in vivo retinal ischemia-reperfusion model and relevant readouts demonstrated by this study will certainly inspire further investigation of the dynamic morphological and functional changes of retinal neurons and glial cell responses during disease progression and before and after treatments.

Weaknesses:

The revised manuscript has been significantly improved in clarity and readability. It has addressed all my questions convincingly.

---

## [Referee Report · Reviewer #2 (Public Review)]

Summary:

The authors of this manuscript aim to develop a novel animal model to accurately simulate the retinal ischemic process in retinal artery occlusion (RAO). A unilateral pterygopalatine ophthalmic artery occlusion (UPOAO) mouse model was established using silicone wire embolization combined with carotid artery ligation. This manuscript provided data to show the changes of major classes of retinal neural cells and visual dysfunction following various durations of ischemia (30 minutes and 60 minutes) and reperfusion (3 days and 7 days) after UPOAO. Additionally, transcriptomics was utilized to investigate the transcriptional changes and elucidate changes in the pathophysiological process in the UPOAO model post-ischemia and reperfusion. Furthermore, the authors compared transcriptomic differences between the UPOAO model and other retinal ischemic-reperfusion models, including HIOP and UCCAO, and revealed unique pathological processes.

Strengths:

The UPOAO model represents a novel approach for studying retinal artery occlusion. The study is very comprehensive.

Weaknesses:

Originally, some statements were incorrect and confusing. However, the authors have made clarifications in the revised manuscript to avoid confusion.

---

## [Author Response]

The following is the authors’ response to the current reviews.

**eLife assessment:**
The manuscript establishes a sophisticated mouse model for acute retinal artery occlusion (RAO) by combining unilateral pterygopalatine ophthalmic artery occlusion (UPOAO) with a silicone wire embolus and carotid artery ligation, generating ischemia-reperfusion injury upon removal of the embolus. This clinically relevant model is useful for studying the cellular and molecular mechanisms of RAO. The data overall are solid, presenting a novel tool for screening pathogenic genes and promoting further therapeutic research in RAO.

Thank you for your thorough evaluation. We are pleased that you find our mouse model for acute retinal artery occlusion to be sophisticated and clinically relevant. Your recognition of the model’s utility in studying the cellular and molecular mechanisms of RAO, as well as its potential for advancing therapeutic research, is highly encouraging and underscores the significance of our work. We are grateful for your supportive feedback.

**Public Reviews:**

**Reviewer #1:**
Summary:Wang, Y. et al. used a silicone wire embolus to definitively and acutely clot the pterygopalatine ophthalmic artery in addition to carotid artery ligation to completely block blood supply to the mouse inner retina, which mimic clinical acute retinal artery occlusion. A detailed characterization of this mouse model determined the time course of inner retina degeneration and associated functional deficits, which closely mimic human patients. Whole retina transcriptome profiling and comparison revealed distinct features associated with ischemia, reperfusion, and different model mechanisms. Interestingly and importantly, this team found a sequential event including reperfusion-induced leukocyte infiltration from blood vessels, residual microglial activation, and neuroinflammation that may lead to neuronal cell death.Strengths:Clear demonstration of the surgery procedure with informative illustrations, images, and superb surgical videos.Two time points of ischemia and reperfusion were studied with convincing histological and in vivo data to demonstrate the time course of various changes in retinal neuronal cell survivals, ERG functions, and inner/outer retina thickness.The transcriptome comparison among different retinal artery occlusion models provides informative evidence to differentiate these models.The potential applications of the in vivo retinal ischemia-reperfusion model and relevant readouts demonstrated by this study will certainly inspire further investigation of the dynamic morphological and functional changes of retinal neurons and glial cell responses during disease progression and before and after treatments.

We sincerely appreciate your detailed and positive feedback. These evaluations are invaluable in highlighting the significance and impact of our work. Thank you for your thoughtful and supportive review.

Weaknesses:The revised manuscript has been significantly improved in clarity and readability. It has addressed all my questions convincingly.

Thank you for your positive feedback. We are pleased to hear that the revisions have significantly improved the manuscript's clarity and readability, and that we have convincingly addressed all your questions. Your encouraging words are of great importance to us.

**Reviewer #2 (Public Review):**
Summary:The authors of this manuscript aim to develop a novel animal model to accurately simulate the retinal ischemic process in retinal artery occlusion (RAO). A unilateral pterygopalatine ophthalmic artery occlusion (UPOAO) mouse model was established using silicone wire embolization combined with carotid artery ligation. This manuscript provided data to show the changes of major classes of retinal neural cells and visual dysfunction following various durations of ischemia (30 minutes and 60 minutes) and reperfusion (3 days and 7 days) after UPOAO. Additionally, transcriptomics was utilized to investigate the transcriptional changes and elucidate changes in the pathophysiological process in the UPOAO model post-ischemia and reperfusion. Furthermore, the authors compared transcriptomic differences between the UPOAO model and other retinal ischemic-reperfusion models, including HIOP and UCCAO, and revealed unique pathological processes.Strengths:The UPOAO model represents a novel approach for studying retinal artery occlusion. The study is very comprehensive.

Thank you for your positive feedback. We are delighted that you find the UPOAO model to be a novel and comprehensive approach to studying retinal artery occlusion. Your recognition of the depth and significance of our study is highly valuable and encourages us in our ongoing research.

Weaknesses:Originally, some statements were incorrect and confusing. However, the authors have made clarifications in the revised manuscript to avoid confusion.

We sincerely appreciate your meticulous review of the manuscript. We have thoroughly addressed the inaccuracies identified in the revised version. Additionally, we have polished the article to ensure improved readability. We apologize for any confusion caused by these inaccuracies and genuinely. We appreciate your careful attention to detail, and your patience and meticulous suggestions have significantly improved the clarity and readability of our manuscript.

The following is the authors’ response to the original reviews.

**Recommendations for the authors:**

**Reviewer #1:**
The revised manuscript has been significantly improved in clarity and readability. It has addressed all my questions convincingly.

Thank you for your positive feedback. We are pleased to hear that the revisions have significantly improved the manuscript's clarity and readability, and that we have convincingly addressed all your questions. Your encouraging words are of great importance to us.

**Reviewer #2:**
The authors have revised the manuscript and/or provided answers to the majority of prior comments, which have helped to strengthen the work. However, addressing the following concerns is still necessary to further improve the manuscript.

Thank you for acknowledging our revisions and the improvements made to the manuscript. We appreciate your continued feedback and will address the remaining concerns to further enhance the quality of our work.

The quantification method of RGCs is described in detail in the response letter, but this detailed methodology was not included in the revised manuscript to clarify the quantification process.

Thank you for your helpful recommendations. We have added detailed methodology in the revised manuscript to clarify the quantification process (line 180-188).

The graphs in Fig. 3D b-wave and Fig. 3E-b wave are duplicated.

We apologize for the error in our figures. We have corrected the mistake by replacing the duplicated image in Fig. 3E-b wave with the correct one (line 880). Your careful observation has been very helpful in improving our manuscript. Thank you for bringing this to our attention.

The quantifications of the thickness of retinal layers in HE-stained sections in Figure 4 (IPL) and Response Figure 2 are incorrect. For mice retina, the thickness of the IPL is approximately 50 µm.

Thank you for your meticulous review of the manuscript. We have rectified the inaccuracies in the quantification of retinal layer thickness in HE-stained sections in Figure 4, addressing the initial issue with the scale bar.

We consulted with a microscope engineer and used a microscope microscale to calibrate the scale of the fluorescence microscope (BX63; Olympus, Tokyo, Japan) at the suggestion of the engineer.

We recount the thickness of all layers of the HE-stained retinal section (line 902). The inner retina thickness in Figure 4 has been adjusted under a new scale bar, and the thickness of the outer retinal layers is now displayed in

Author response image 1. However, the IPL thickness of the sham eye in the UPOAO model is still not aligned with the common thickness of 50 µm. Therefore we review the literature within our laboratory, focusing on C57BL/6 mice from the same source, revealed that the inner retina thickness (GCC+INL) in the HE-stained sections of the sham eye in the UPOAO model (around 80 µm) is consistent with previous findings (see Author response image 2) conducted by Kaibao Ji and published in *Experimental Eye Research* in 2021 [1].

We captured and analyzed the average retinal thickness of each layer over a long range of 200-1100 μm from the optic nerve head (see Author response image 3, highlighted by the green line). The field region has been corrected in the revised manuscript (line 232). Considering the significant variation in retinal thickness from the optic nerve to the periphery, we consulted literature on multi-point measurements of HE-stained retinas. The average thickness of the GCC layer in the control group was approximately 57 µm at 600 µm from the optic nerve head and about 48 µm at 1200 µm from the optic nerve head in the literature [2] (see Author response image 4). The GCC layer thickness of the sham eye in the UPOAO model is around 50 µm, in alignment with existing literature. In future studies, we will pay more attention to the issue of thickness averaging.

We appreciate your thorough review and valuable feedback, which has enabled us to correct errors and enhance the accuracy of our research.

**Author response image 1. sa3fig1:** Thickness of OPL, ONL, IS/OS+RPE in HE staining. n=3; ns: no significance (p>0.05).

**Author response image 2. sa3fig2:** Cited from Ji, K., et al., Resveratrol attenuates retinal ganglion cell loss in a mouse model of retinal ischemia reperfusion injury via multiple pathways. Experimental Eye Research, 2021. 209: p. 108683.

**Author response image 3. sa3fig3:** Schematic diagram illustrating the selection of regions. The figure was captured using a fluorescence microscope (BX63; Olympus, Tokyo, Japan) under a 4X objective. Scale bar=500 µm.

**Author response image 4. sa3fig4:** Cited from Feng, L., et al., Ripa-56 protects retinal ganglion cells in glutamate-induced retinal excitotoxic model of glaucoma. Sci Rep, 2024. 14(1): p. 3834.

There are some typos in the summary table. For example: 'Amplitudes of a-wave (0.3, 2.0, and 10.0 cd.s/m²)' should be 'Amplitudes of a-wave (0.3, 3.0, and 10.0 cd.s/m²)'; and 'IINL thickness' in HE' should be 'INL thickness'.

Thank you for pointing out the typos in the summary table (line 1073). We have corrected 'Amplitudes of a-wave (0.3, 2.0, and 10.0 cd.s/m²)' to 'Amplitudes of a-wave (0.3, 3.0, and 10.0 cd.s/m²)' and 'IINL thickness' to 'INL thickness'. Your attention to detail is greatly appreciated and has been very helpful in improving our manuscript.

References

(1) Ji, K., et al., Resveratrol attenuates retinal ganglion cell loss in a mouse model of retinal ischemia reperfusion injury via multiple pathways. Experimental Eye Research, 2021. **209**: p. 108683.

(2) Feng, L., et al., Ripa-56 protects retinal ganglion cells in glutamate-induced retinal excitotoxic model of glaucoma. Sci Rep, 2024. **14**(1): p. 3834.